# Paramutation at the maize *pl1* locus is associated with RdDM activity at distal tandem repeats

Natalie C. Deans[1,2¤a], Joy-El R. B. Talbot[1,3¤b], Mowei Li[1,2], Cristian Sáez-González[1¤c], Iris Hövel[4], Darren Heavens[5], Maike Stam[4], Jay B. Hollick[1,2,6]*

1 Department of Molecular Genetics, The Ohio State University, Columbus, Ohio, United States of America, 2 Centers for Applied Plant Sciences and RNA Biology, The Ohio State University, Columbus, Ohio, United States of America, 3 Department of Molecular and Cell Biology, University of California Berkeley, Berkeley, California, United States of America, 4 Swammerdam Institute for Life Sciences, Universiteit van Amsterdam, Amsterdam, The Netherlands, 5 Earlham Institute, Norwich, United Kingdom, 6 Department of Plant and Microbial Biology, University of California, Berkeley, California, United States of America

¤a Current address: Center for Applied Genetic Technologies, University of Georgia, Athens, Georgia, United States of America
¤b Current address: The Roux Institute, Northeastern University, Portland, Maine, United States of America
¤c Current address: Department of Cell Biology, Johns Hopkins University, Baltimore, Maryland, United States of America
* hollick.3@osu.edu

**Data Availability Statement:** Sequence data from this article can be found in the GenBank, SRA, and GEO data libraries under the following accessions, Pl1-Rhoades haplotype sequence (Genbank

## Abstract

Exceptions to Mendelian inheritance often highlight novel chromosomal behaviors. The maize *Pl1-Rhoades* allele conferring plant pigmentation can display inheritance patterns deviating from Mendelian expectations in a behavior known as paramutation. However, the chromosome features mediating such exceptions remain unknown. Here we show that small RNA production reflecting RNA polymerase IV function within a distal downstream set of five tandem repeats is coincident with meiotically-heritable repression of the *Pl1-Rhoades* transcription unit. A related *pl1* haplotype with three, but not one with two, repeat units also displays the *trans*-homolog silencing typifying paramutations. 4C interactions, CHD3a-dependent small RNA profiles, nuclease sensitivity, and polyadenylated RNA levels highlight a repeat subregion having regulatory potential. Our comparative and mutant analyses show that transcriptional repression of *Pl1-Rhoades* correlates with 24-nucleotide RNA production and cytosine methylation at this subregion indicating the action of a specific DNA-dependent RNA polymerase complex. These findings support a working model in which *pl1* paramutation depends on *trans*-chromosomal RNA-directed DNA methylation operating at a discrete *cis*-linked and copy-number-dependent transcriptional regulatory element.

## Author summary

Although switching genes "on" and "off" in a coordinated fashion is essential to organismal development and homeostasis, some genes - or specific forms thereof - display

OQ405024), 4C libraries (SRA SRR23352217-SRR23352222), Pl-Rh, Pl´, and Pl-Rh / Pl´ cob and seedling sRNA libraries (SRA SRR15400896-SRR15400912), B73 sRNAs (GEO GSE52103), rpd1-1 and rmr1-1 sRNAs (SRA SRR15410418-SRR15410422), and dcl3-2 sRNAs (SRA SRR16938685-SRR16938686). All other relevant data are within the manuscript and its Supporting Information files.

**Funding:** Research activities were funded by awards to J.B.H. from the National Research Initiative of the USDA Cooperative State Research, Education and Extension Service (www.usda.gov; 99-35301-7753, 2001-35301-10641, and 2006-35304-17399), the National Science Foundation (www.nsf.gov; MCB-0419909, -0920623, -1715375) and The Ohio State Foundation (www.osu.edu). I.H. was supported by funding of the Research Priority Area Systems Biology of the University of Amsterdam (www.uva.nl). This study and N.C.D. research activity was supported in part by resources and technical expertise from the Georgia Advanced Computing Resource Center (www.gacrc.uga.edu), a partnership between the University of Georgia's Office of the Vice President for research and Office of the Vice President for Information Technology. Additional support of research activities were provided by the Comprehensive Cancer Center and the National Institutes of Health (www.nih.gov) under grant number P30 CA016058 The views expressed are solely those of the authors and are not endorsed by the sponsors of this work. The sponsors and funders did not play any role in the study design, data collection and analysis, decision to publish, or preparation of the manuscript.

**Competing interests:** I have read the journal's policy and the authors of this manuscript have the following competing interests: All rmr materials and their uses are covered by U.S. patent 8134047 assigned to The Regents of the University of California. The authors claim no other known competing interests.

unusual switching behaviors where the "off" state is sexually transmitted to offspring. Further, only "off" states persist when both "on" and "off" states are combined in heterozygous condition. Examples of this unusual behavior, known as paramutation, exist for certain transgenes and/or endogenous genes in both plants and animals but the chromosome features triggering this switching and the responsible cellular machinery are poorly understood. Here, we used long-read DNA sequencing, comparative genomics, DNA -DNA interaction assays, mutant analyses, and cytosine methylation profiling to identify a switchable regulatory sequence associated with paramutation at the maize *purple plant1* (*pl1*) locus. These key sequences are embedded within a 5-copy tandem repeat array of mostly transposable elements ~14 kilobases distal to the *pl1* gene. The "off" state is associated with these sequences producing so-called "small interfering RNAs" (siRNAs) that may direct the recruitment of chromatin-modifying enzymes. Our findings support a molecular model for paramutation and highlight the regulatory importance of repeated sequences found in large eukaryotic genomes.

## Introduction

Patterns of transcriptional regulation governing normal development are consistently replicated from one generation to the next. However, certain *trans*-homolog interactions known as paramutations facilitate heritable changes in regulation–canonically observed as an active allele being heritably suppressed upon exposure to a repressed allele in *trans* (reviewed in [1,2]). Susceptible alleles are "paramutable" while repressed derivatives have "paramutagenic" potential. Despite the potential evolutionary significance of such behaviors, the chromosome organization and nuclear feature(s) that define these dynamic properties remain largely unknown.

Paramutation-like behaviors have been described for endogenous alleles in several species including maize (*Zea mays*) [3–7], tomatoes (*Solanum lycopersicum*) [8], Arabidopsis (*Arabidopsis thaliana*) [9], mice (*Mus musculus*) [10], and potentially humans [11]. Specific transgenes displaying paramutation-like behaviors are also documented in petunia [12], tobacco (*Nicotiana tobaccum*) [13], Arabidopsis [14,15], mice [16,17], flies (*Drosophila melanogaster*) [18,19], and worms (*Caenorhabditis elegans*) [20–24]. Many of these pan-eukarya examples implicate roles for small RNA (sRNA)-based chromatin modifications (reviewed in [25]) highlighting an ancient mechanism of genome control having potential for the transgenerational inheritance of novel regulatory programs [26].

In maize, the first paramutation examples were reported for specific *red1/colored1* (*r1*) alleles [3] which encode basic helix-loop-helix (bHLH) transcription factors conferring seed and plant color [27,28] and an allele of the paralogous *booster1/colored plant1* (*b1*) locus [4]. Other alleles exhibiting paramutation are found at the *purple plant1* (*pl1*) [6] and *pericarp color1* (*p1*) [5] loci that encode R2R3 MYB-type transcription factors required for plant and cob pigmentation, respectively [28]. An additional example occurring at the *low phytic acid1* (*lpa1*) locus [7] shows that paramutations in maize are not confined to genes encoding transcriptional regulators of pigment production.

Forward genetic screens for *mediators of paramutation* (*mop*) of the *B1-Intense* (*B1-I*) allele [29,30] and factors *required to maintain repression* (*rmr*) of the *Pl1-Rhoades* allele [31,32] identify both overlapping and locus-specific functions. Several *mop* and *rmr* loci encode orthologs of proteins involved in an Arabidopsis RNA-directed DNA methylation (RdDM) pathway. RdDM utilizes two plant-specific DNA-dependent RNA polymerases (RNAPs IV and V) and

accessory proteins to guide *de novo* cytosine methylation. RNAP IV indirectly produces mostly 24 nucleotide (24nt) RNAs that, in complex with Argonaute proteins, help recruit a *de novo* cytosine methyltransferase to nascent long-non-coding RNAP V transcripts (reviewed in [33]). *rmr6*/*mop3* / *rpd1* encodes the maize RNAP IV largest subunit, RPD1 [34,35], and *rmr7*/ *mop2* encodes one of at least two second largest subunits shared between RNAPs IV and V, RP (D/E)2a [30,36,37]. RPD1 and RP(D/E)2a are orthologous to the Arabidopsis NUCLEAR RNA POLYMERASE D1 (NRPD1) and NRP(D/E)2, respectively. Maize has three genes producing second largest subunits that, according to proteomic data, constitute distinct RNAP IV and RNAP V subtypes [37]. *mop1* encodes an ortholog of RNA-DEPENDENT RNA POLY-MERASE2 (RDR2) [38] that converts nascent RNAP IV transcripts to a double-stranded substrate for a dicer-like3 (DCL3) endonuclease that generates primarily 24nt RNAs. *rmr5* encodes the sole maize DCL3 [39].

The other three RMR proteins identified to date, (RMR1, RMR2, and RMR12/CHD3a), also influence 24nt RNA biogenesis, but neither RMR1 nor RMR12/CHD3a are required to mediate *b1* paramutation or to maintain repressed *B1-I* states - denoted *B´* [32,40]. RMR1 is a likely SNF2-type ATPase, potentially orthologous to Arabidopsis CLASSY3 and 4 [32], that co-immunoprecipitates with RPD1 in undifferentiated callus cells [37]. RMR2 is a pioneer protein whose potential Arabidopsis orthologs have no known function [41], and RMR12/ CHD3a is a chromodomain helicase DNA-binding3 (CHD3) protein orthologous to the Arabidopsis nucleosome-remodeling protein, PICKLE [40]. Both PICKLE and RMR12/CHD3a influence genome-wide distributions of 24nt RNA production [40,42].

Additional proteins implicated in the repression of *B´* include a putative maize ortholog of Arabidopsis ARGONAUTE9 (AGO9) (a member of the AGO4 clade), ZmAGO104, [43] and a novel protein isoform encoded by the *unstable factor for orange1* (*ufo1*) locus [44,45]. A dominant *Ufo1-1* allele enhances the expression of both repressed *B´* and *p1* alleles through an unknown mechanism [44].

One working model for paramutation in maize envisions 24nt RNAs produced from a paramutagenic allele serving as a diffusible substance directing cytosine methylation and/or repressive histone modifications to an active paramutable allele leading to heritable repression. Chromatin modifications, including 5-methylcytosine (5mC), or the 24nt RNAs themselves, may serve as a meiotically heritable mark of paramutagenic alleles [2,25,46]. This model is supported by 1) the associations of MOP and RMR proteins with 24nt RNA biogenesis, 2) transgenic shRNA-mediated induction of *b1* paramutation [47], 3) increased 5mC levels at paramutagenic *r1* [48], *p1* [44,49,50], and *b1* alleles [51], and 4) transcriptional repression of paramutagenic *B1-I* [52], *P1-rr* [53], and *Pl1-Rhoades* [54] alleles. However, paramutations at *pl1* can still occur with reduced frequencies in *rmr1-1/rmr1-2* and *rmr2-1* mutants [32,41], paramutation at *b1* still occurs in a parent-of-origin specific manner in the *rmr7-1* mutant [36], sRNA profiles of active (*B-I*) and repressed (*B´*) *B1-I* states appear similar [47], no 5mC differences have so far been found between active (*Pl-Rh*) and repressed (*Pl´*) *Pl1-Rhoades* states [32,54], and–with the possible exception of RP(D/E)2a –no known MOP or RMR proteins represent RdDM-type components acting downstream of 24nt biogenesis. These findings contrast the expectations of a canonical RdDM-like model.

In maize, paramutagenic haplotypes are associated with repetitive sequences. At *r1*, the repeats include entire *r1*-coding regions plus flanking intergenic sequences [55–57], while *B1-I* and *P1-rr* have intergenic repeats acting as transcriptional enhancers [58,59]. A hepta-repeat of an otherwise unique 853bp sequence found ~100kb 5´ of the *B1-I* coding sequence (CDS) acts as both an enhancer and an efficient inducer of *B1-I* paramutation [59,60]. These findings indicate that the *b1* hepta-repeat can both promote expression of *B-I* and confer

paramutagenic activity to *B´*. At both *r1* and *b1*, repeat numbers directly correlate with para-mutagenic strength [57,59–61] but the nature of this relationship remains speculative [62].

Although the sequences mediating paramutation at *Pl1-Rhoades* have remained unknown, a *Pl1-Rhoades || pl1-B73* recombinant derivative, *pl1-R30*, defines a region > 6kb 3´ of the *pl1* CDS required for conferring strong pigmentation and for paramutagenicity [63]. *pl1-R30* contains the *Pl1-Rhoades* CDS and a promoter-proximal CACTA-type DNA transposable element (TE) fragment of the *doppia* subclass [56,63,64] but confers uniformly weak pigmentation similar to that of *pl1-B73*. 5mC patterns at this *doppia* fragment are dependent on both RPD1 and RMR1 [32] but appear similar in both *Pl-Rh* and *Pl´* states [54]. This TE fragment likely directs seed-specific expression because 1) a similar *doppia* fragment appears to serve as a seed-specific promoter in the paramutable *R-r:standard* haplotype [56]; 2) the non-paramutable *Pl1-Blotched* (*Pl1-Bh*) allele that has near-identical sequence to *Pl1-Rhoades* over at least 3.5kb, including the *doppia* sequences [65,66], confers blotchy seed pigmentation [65]; and 3) *Pl1-Rhoades* becomes seed expressed after multi-generational absence of RPD1 [63]. Thus, RNAP IV is responsible for the tissue-restricted expression of *Pl1-Rhoades*, and other genes [67,68], potentially by repressing regulatory TEs [68,69]. Importantly, the non-paramutable nature of both *pl1-R30* and *Pl1-Bh* indicate the *doppia* fragment by itself is insufficient for paramutagenic function.

Here we document the *Pl1-Rhoades* haplotype structure and molecular changes occurring within a distal tandem repeat that are associated with paramutation. A 192kb assembly of long-read sequences of bacterial artificial chromosomes (BACs) containing the *Pl1-Rhoades* CDS revealed a 2092bp sequence present in five tandem direct copies beginning ~14kb 3´ of the *pl1* polyadenylation site (PAS). Structure comparisons indicate that three repeat copies are present in another paramutable *pl1* haplotype from the CML52 inbred. Circular chromosome conformation capture (4C) results show these repeats are in close proximity to a bait sequence located between the 5´ *doppia* fragment and *Pl1-Rhoades* transcription start site (TSS). One or more of these repeats differ in both nuclease sensitivity and polyadenylated RNA levels between active *Pl-Rh* and repressed *Pl´* states. We also show that an otherwise unique portion of the repeat unit produces RNAP IV-dependent 24nt RNAs only in the *Pl´* state, identifying a ~200bp segment as having differential RNAP IV recruitment between *Pl-Rh* and *Pl´*. We found that this precise positioning of sRNA biogenesis is defined by CHD3a. In contrast, no significant sRNA differences were observed at the *doppia* fragment or elsewhere in the 192kb haplotype. The differential sRNA abundances also coincide with differing 5mC patterns. Our findings are consistent with a model in which *Pl1-Rhoades* paramutation involves a maize-specific RNA-directed DNA methylation-like mechanism acting at a copy-number-dependent distal *cis*-regulatory element.

## Results

### The *Pl1-Rhoades* haplotype contains a penta-repeat

To extend the existing *Pl1-Rhoades* reference sequence [40], six *Pl1-Rhoades*-containing BAC clones were sequenced, and resulting long-reads were assembled into a 192kb contig (Fig 1 and see S1 Table; Genbank OQ405024). Comparison to the B73 AGPv4 reference genome [70] highlighted two major structural variations downstream of the *Pl1-Rhoades* CDS, including the absence of a B73-specific gene model (Zm00001d037119) and the presence of five tandem repeats, each having an identical 2092bp sequence, located between ~14kb 3´ of the PAS and ~100bp 3´ of the next, convergent gene model (Zm00001d037120)(Fig 1D). A CENSOR-based annotation of the repeated sequence [71] identified mostly TE fragments with two small DNA TEs (one highly repetitive DNA-7_ZM and one relatively unique PIF-Harbinger-type)

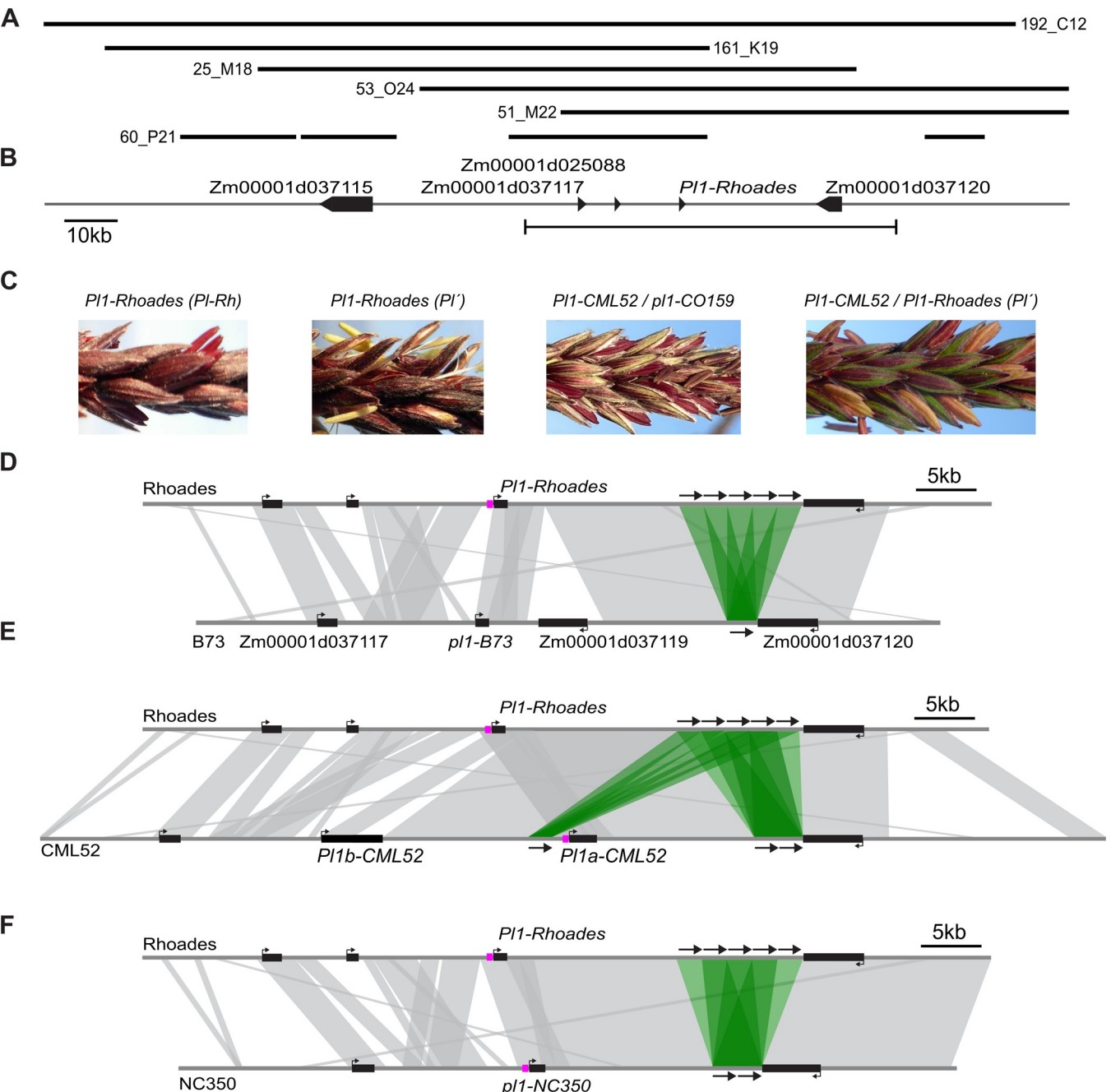

**Fig 1. *pl1* haplotypes. (A)** BAC clone contigs containing *Pl1-Rhoades* assembled from long-read sequences. **(B)** *Pl1-Rhoades* haplotype representation with arrows identifying annotated B73 AGPv4 gene models. Bracketed line represents the region used for *GEvo* alignments with *pl1-B73* **(D)**, *Pl1-CML52* **(E)**, and *pl1-NC350* **(F)**. **(C)** Images of anther phenotypes conferred by the respective active and repressed states of the *Pl1-Rhoades* and *Pl1-CML52* alleles. *pl1-CO159* is a recessive non-functional allele used here to illustrate the pigmentation potential of *Pl1-CML52*. **(D-F)** Shaded areas represent regions of synteny, black boxes represent gene models, and magenta boxes immediately upstream of the *pl1* gene models represent a *doppia* transposon fragment. Green shading and black arrows represent sequences found as a penta-repeat in *Pl1-Rhoades*.

flanking a 390bp unique non-coding region hereafter referred to as the unique subregion (USR)(see S1A Fig; see S1 File; 135,133–135,522 in Genbank sequence OQ405024). Southern blot hybridizations (see S1B–S1D Fig) with a radiolabeled USR probe showed that both

*pl1-B73* and *pl1-R30* have single similar-sized *Bgl*II fragments but no ~2kb fragments diagnostic of the tandem repeats. Additionally, the ~4kb *pl1-B73* and *pl1-R30 Nsi*I fragments contrast the >12kb *Pl1-Rhoades* fragment encompassing the five repeats, hereafter referred to as the penta-repeat. Together with previous results [63], these data indicate the *pl1-R30* recombination event occurred between ~6 and 12kb 3′ of the *Pl1-Rhoades* PAS (see S1B–S1C Fig). Importantly, recombinant loss of the penta-repeat in *pl1-R30* is coincident with the absence of both high *pl1* gene expression and paramutagenicity [63].

## Repeat copy numbers correlate with paramutagenicity

So far, the only *pl1* alleles shown to have or acquire paramutagenic activities are *Pl1-Rhoades* and *Pl1-CML52* [63,66], the latter being present in one of the 25 founder lines (CML52) of a nested association mapping (NAM) population [72]. Both alleles can condition strong pigmentation of anthers and plant body ([6]; Fig 1C). We queried all founder line long-read assemblies [73] with the USR sequence and found all contain >80% of this sequence with >97% identity, but only the CML52 and NC350 lines contain more than a single intact repeat unit. The CML52 haplotype consists of two *pl1* CDSs (Fig 1E) distinguished by a 6bp deletion and three 3bp insertions unique to *Pl1b-CML52*, and four synonymous and two conservative non-synonymous bp substitutions. The 5′-most gene model (*Pl1b-CML52)* also has a large annotated 3′ UTR intron, while the other (*Pl1a-CML52*) is identical to *Pl1-Rhoades*, aside from a 32bp insertion into the adjacent *doppia* fragment [63]. Two tandem copies of the repeat are found precisely where the five are located in *Pl1-Rhoades* and a third solo copy lies just upstream of *Pl1a-CML52* (Fig 1E). Both the solo repeat unit and the first of the downstream tandem repeats are identical to those in *Pl1-Rhoades* while the second downstream repeat unit has a SNP.

The NC350 *pl1* haplotype is also similar to *Pl1-Rhoades*, but only has two repeats (Fig 1F). The CML52, NC350 and *Pl1-Rhoades* haplotypes are nearly identical over ~20kb encompassing the *doppia* fragment and first two tandem repeats. Differences include a CML52- and NC350-shared 32bp *doppia* insertion, 4 CML52-specific SNPs, a single NC350-specific SNP within the *doppia* fragment, one *Pl1-Rhoades*-specific SNP, and an intergenic poly-T sequence with lengths of 8, 9, and 10nt in *Pl1-Rhoades*, NC350, and CML52, respectively. Given the structural similarity of the NC350 haplotype with others having paramutagenic potential, we used an established pedigree analysis with a linked genetic marker [74] to test whether it could acquire paramutagenic activity following exposure to *Pl′* (see S1 Methods). With one possible recombinant exception, no paramutagenic activity co-segregated with the NC350 *pl1* haplotype (see S1 Methods and S2 Table). Similar tests showed that none of seven other NAM founder line *pl1* haplotypes containing single repeat units acquired paramutagenic activity (see S1 Methods and S2 Table). Because the CML52 *pl1* haplotype becomes paramutagenic following exposure to *Pl′* [63], we infer that haplotypes having more than two of these repeats have paramutation-like behaviors. Additionally, paramutagenicity of the CML52 haplotype is less than that of *Pl1-Rhoades* (see S2 Table; [63]) consistent with repeat copy numbers correlating with paramutagenic strength.

## Repeat sequences associate with the *Pl1-Rhoades* promoter

The extended sequence of the *Pl1-Rhoades* haplotype enabled a 4C analysis to identify potential distal enhancers. Using a bait sequence located 20nt 5′ of the *Pl1-Rhoades* TSS, we generated and sequenced 4C libraries of *Dpn*II - *Nla*III fragments derived from nuclei of both inner seedling leaves (S) and inner husk leaves (H) harvested from *Pl-Rh* / *Pl-Rh* plants (see S3 Table). In both tissues *pl1* mRNA is highly expressed relative to *Pl′* / *Pl′* genotypes (see

S2 Fig; [54]). Sequence tags were first mapped to the *Pl1-Rhoades* haplotype with the penta-repeat collapsed to a single unit and ranked by abundance for each library, excluding those within 2kb of the bait sequence (see S4 Table). The highest tag density overlapped the repeat sequence, with three of the five most abundant H tags and two of the top five S tags residing in the repeat region. Mapping all 4C tags to the B73 AGPv4 genome indicated that many of the most abundant tags represented highly repeated sequences. We therefore repeated the 4C tag analysis by 1) excluding reads multiply-mapping to the genome, 2) mapping this filtered set to the *Pl1-Rhoades* haplotype, 3) ranking their abundances (see S5 Table), and 4) displaying these in 2kb bins (Fig 2). The highest tag density again overlapped the repeat sequence with the most abundant tags from both tissues (see S1 and H1) located within the USR (Fig 2E). These data indicate that the penta-repeat is physically close to the *Pl1-Rhoades* promoter in multiple tissues of *Pl-Rh* plants in which *pl1* mRNA is abundant, including inner husk leaves where the gene is highly transcribed [54].

## The penta-repeat is targeted by CHD3a

CHD3a specifies genome-wide locations of sRNA biogenesis and is required to maintain both somatic and meiotic repression of *Pl′* states [40]. To further investigate the potential

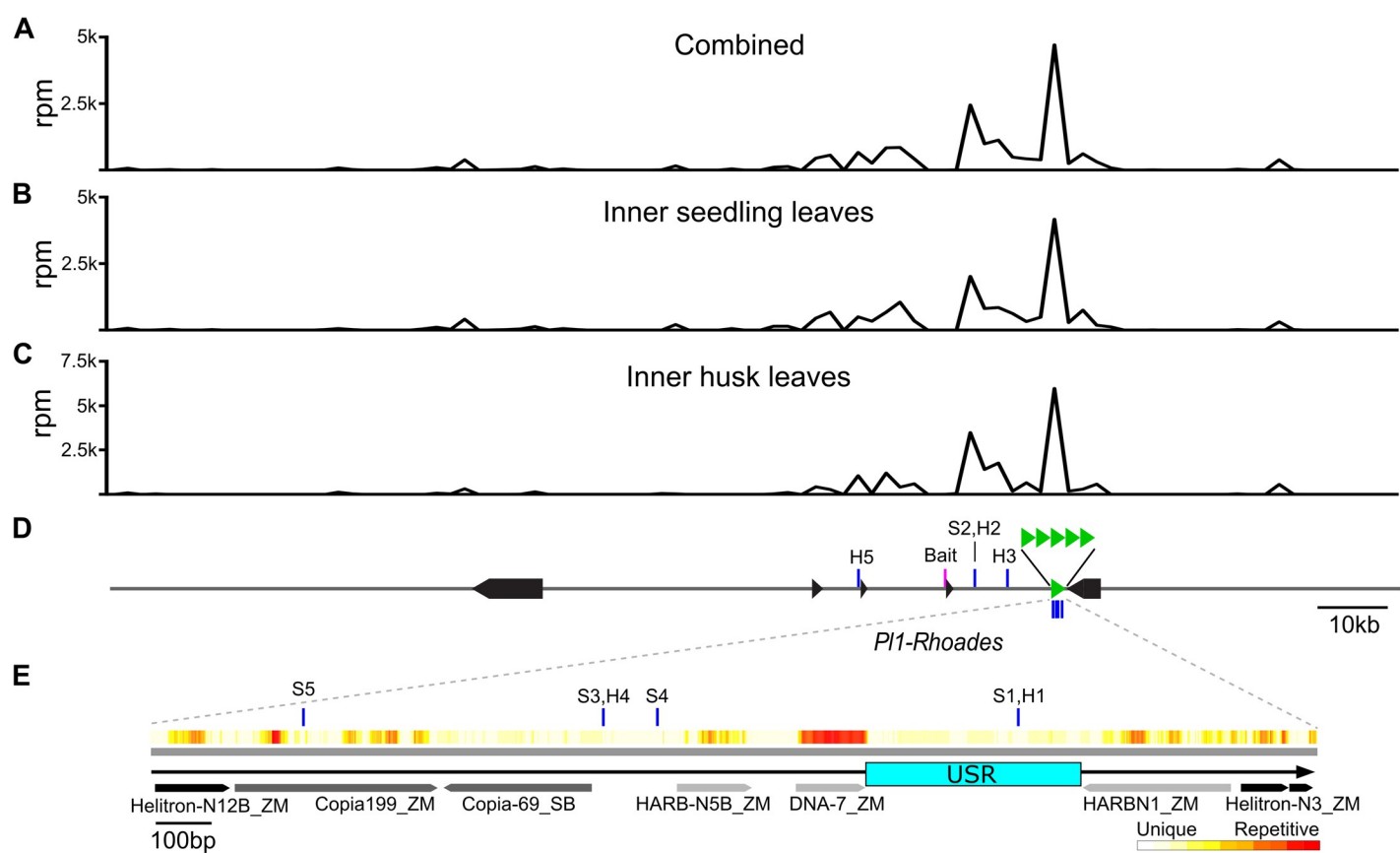

**Fig 2. *Pl1-Rhoades* haplotype 4C-based interactions.** Uniquely-mapping reads per million (rpm) 4C tags in 2kb bins across the *Pl1-Rhoades* haplotype: schematic combined datasets **(A)**, inner seedling leaves (S) **(B)**, and inner husk leaves (H) **(C)**. **(D)** Schematic identifying positions and ranks of the five most abundant 4C tags (blue bars) ligated to the bait sequence (magenta bar) in S and H samples excluding 2kb on either side of the bait. Black arrows represent annotated gene models. A single copy of the penta-repeat (green triangles) is displayed in **(E)** and shows 4C tag positions and ranks (as above), repetitive index (heat map) per 24nt sliding windows, and annotated TE features. Cyan box represents a unique subregion (USR). Arrows represent DNA transposons (light gray), *Helitrons* (black), and LTR retrotransposons (dark gray).

involvement of the penta-repeat in *Pl1-Rhoades* regulation, we aligned existing day-8 seedling-derived 18-30nt RNA reads (excluding those multiply-mapping to the genome; see Methods) from *Pl´ Chd3a* non-mutant and *chd3a-3* mutant siblings to the *Pl1-Rhoades* haplotype and looked for differential sRNA patterns. In *Pl´ Chd3a* profiles, the primary sRNA source across the repeat is within the USR (Fig 3). In both *Chd3a* non-mutants and *chd3a-3* mutants, the USR sRNA reads are primarily 24nt (24mers)(79% and 73% respectively) and these 24mer fractions are not significantly different between genotypes (*p* = 0.27, 2-sample *t*-test). In the *chd3a-3* mutant profiles, however, a 48% reduction (*p* = 0.02, 2-sample *t*-test) of USR sRNAs is observed and this reduction is associated with a novel peak (15.8-fold increase) of sRNAs mapping to the flanking PIF-Harbinger-type TE (*p* = 0.03, 2-sample *t*-test)(Fig 3). Again, 24mers represent the majority of sRNAs mapping to this TE in both mutants (77.7%) and non-mutants (67.8%), and these fractions do not differ significantly between genotypes (*p* = 0.56, 2-sample *t*-test). Because 24nt RNAs are diagnostic of RNAP IV action [34], these findings imply that CHD3a action normally positions RNAP IV at the USR in a paramutagenic *Pl´* state. In the absence of CHD3a, RNAP IV is still recruited to penta-repeat sequences but now acts on a DNA TE adjacent to the USR. These data indicate that one or more penta-repeat units are a target of both RNAP IV and CHD3a actions.

## USR nuclease sensitivity and RNAP II-derived RNA levels differ between *Pl-Rh* and *Pl´* states

We reasoned that because the USR is targeted by CHD3a, differences in chromatin structure might distinguish *Pl-Rh* and *Pl´*. To test this, we measured relative USR nuclease sensitivity by micrococcal nuclease (MNase)-qPCR. After using a mild MNase digestion to produce a gel-separated ladder of nucleosome fragments, mono- and dinucleosome-bound DNA was isolated from pooled isogenic day-8 *Pl-Rh* (*Pl-Rh* / *Pl-Rh*) and *Pl´* (*Pl´* / *Pl´*) seedlings (3 biological replicates each) and the presence of USR sequences was quantified using four tiled amplicons (Fig 4). The central USR region (amplicons P2 and P3) was significantly more associated with

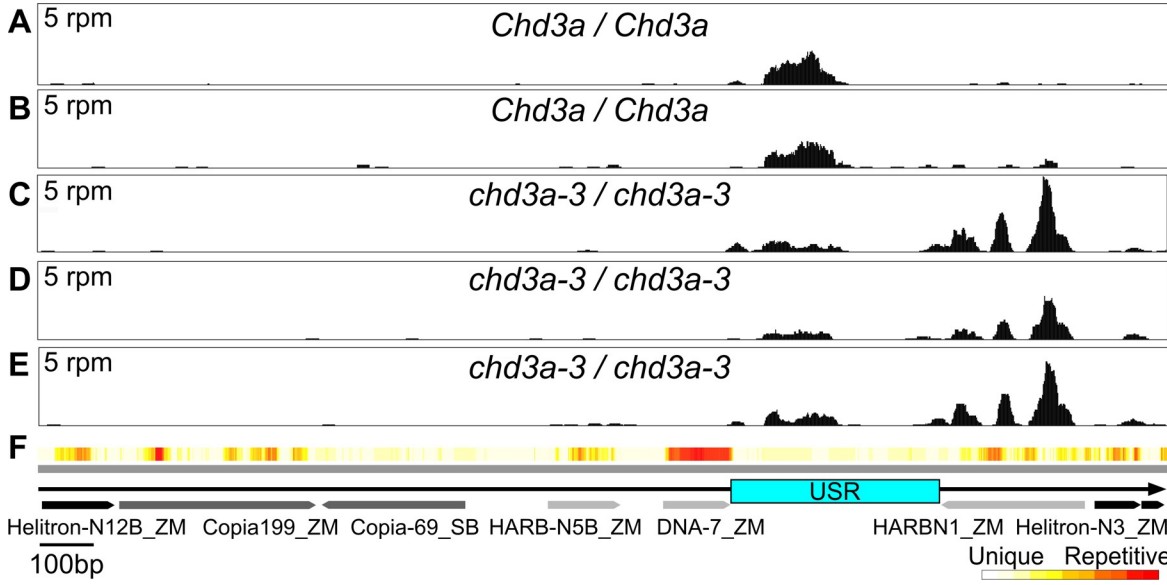

**Fig 3. *Chd3a*-dependent penta-repeat sRNA profiles.** Alignments of uniquely-mapping 18-30nt reads from libraries representing single *Chd3a* / *Chd3a* (**A-B**), and *chd3a-3* / *chd3a-3* (**C-E**) *Pl´* 8-day seedlings across a single repeat unit (**F**) in reads per million (rpm). Arrows represent DNA transposons (light gray), *Helitrons* (black), and LTR retrotransposons (dark gray).

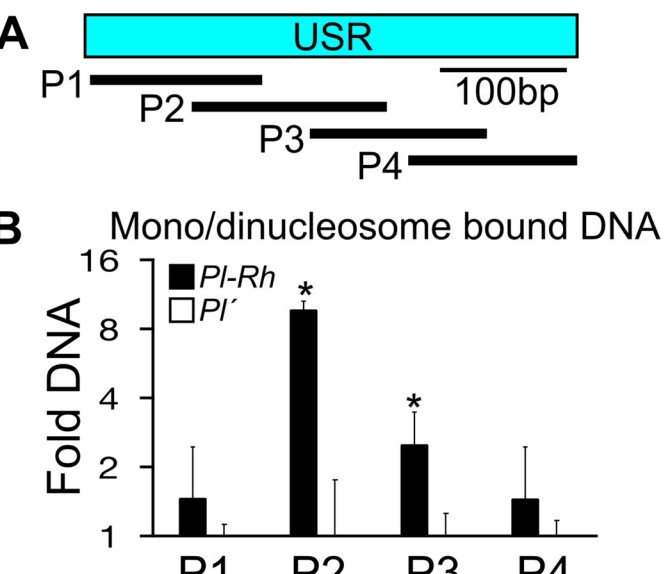

**Fig 4. USR nucleosome profiles.** (A) Relative locations of amplicons (black lines) used in (B). (B) Fold DNA relative to *Pl´* samples of mono- and dinucleosome bound DNA after light MNase digestion measured by qPCR in triplicate *Pl-Rh* and *Pl´* seedlings. * $p < 0.05$. Error bars are s.e.m.

mono- and dinucleosomes in the *Pl-Rh* state compared to *Pl´* ($p<0.05$; two-sample t-test)(Fig 4B). From this result we infer that the *Pl´* USR is either associated with a less accessible and therefore less nuclease sensitive chromatin structure, or it is more nucleosome free and hence more nuclease sensitive than that of *Pl-Rh*. Regardless, changes in USR MNase accessibility in day-8 seedlings are associated with differences in seedling and plant pigmentation [6].

We next evaluated whether *Pl1-Rhoades* expression also corresponded with RNAP II-dependent USR-derived RNA levels using oligo(dT)-primed qRT-PCR on RNA isolated from triplicate isogenic *Pl-Rh* and *Pl´* individuals. We chose male flowers for this analysis since *Pl1-Rhoades*-dependent pigment production is much stronger in male flowers than in day-8 seedlings. Poly A+ USR-containing RNA levels were 15-fold higher ($p = 0.027$, two-sample *t*-test) in *Pl-Rh* versus *Pl´* individuals (Fig 5). Thus, differences in chromatin accessibility at USR sequences producing CHD3a-dependent sRNAs in seedlings coincides with differential USR RNA abundances that positively correlate with *Pl1-Rhoades* expression.

## Paramutagenic states are associated with an RNAP IV complex generating USR 24nt RNAs

All known MOP and RMR factors influence 24nt RNA production, or locations thereof, therefore we looked for paramutation-specific sRNA profiles between biologically triplicate isogenic *Pl-Rh* and *Pl´* individuals. Six combined *Pl-Rh* and *Pl´* 4cm cob sRNA datasets (see S6 Table for library statistics) were first genome filtered to exclude multiply-mapping reads (see Methods) and then used to call uniquely-mapping 18-30nt sRNA clusters across the *Pl1-Rhoades* haplotype using *ShortStack* [75]. Normalized read abundances from each dataset were then compared for each cluster. Thirty clusters were identified–all dominated by 24mers, mean 80%–(see S2 File; see S3 Fig) with clusters 24–25 overlapping the repeat region (24 immediately before the USR and 25 overlapping the USR and adjacent region) and 12–13 covering parts of the *doppia* fragment (see S4 Fig). Only cluster 25 differed significantly, with average sRNA abundances 9.9-fold higher in *Pl´* compared to *Pl-Rh* libraries (see S2 File; $p = 0.003$,

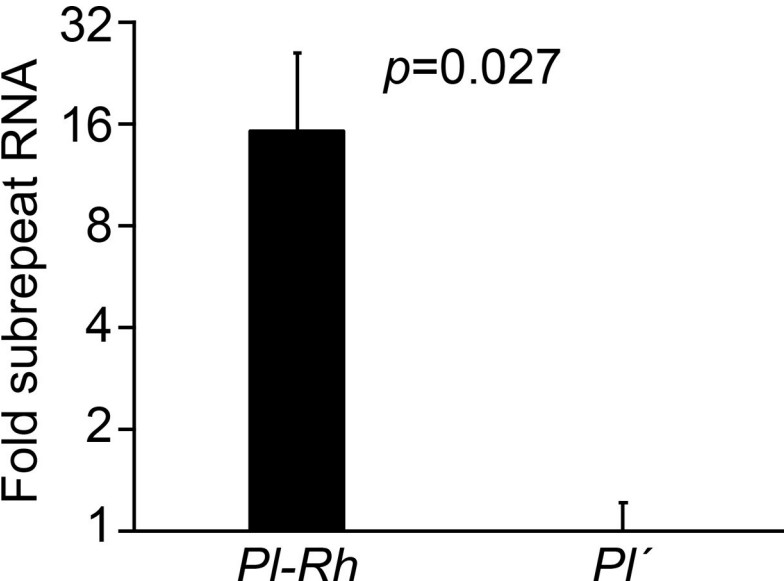

**Fig 5. USR poly A+ RNA levels.** Mean fold RNA ($2^{-\Delta\Delta Ct}$)(±s.e.m) across the USR measured by qRT-PCR normalized to *gapdh* levels in biological triplicate *Pl-Rh* and *Pl´* male flowers. Error bars are s.e.m.

2-sample *t*-test). Day-8 seedling sRNA abundances were also compared across these same 30 clusters (see S2 File) and only cluster 25 was significantly different; 4.4-fold higher in the *Pl´* libraries ($p = 0.002$, 2-sample *t*-test). Cluster 25 also has the most abundant sRNA reads in *Pl´* and *Pl-Rh / Pl´* datasets from both cob and seedling. Thus, in both vegetative and reproductive tissues, the *Pl´* state is associated with increased sRNA production specifically from a subregion of one or more penta-repeat units.

Visualizing both cob and seedling sRNA read alignments, we found abundant USR reads in both *Pl´* and *Pl-Rh / Pl´* heterozygotes and very few of these reads in *Pl-Rh* libraries (see Figs 6, S5 and S6). Thus, the significant differences in cluster 25 read abundances noted above predominantly reflect the USR profiles. To further characterize these USR-derived sRNAs, the reads were collapsed to their 5´-most coordinate and the mean reads per million (rpm) were graphed in 10nt bins for each tissue and genotype (Fig 6H–6M), colored by length and separated by strand. In *Pl´* and *Pl-Rh / Pl´* cobs and seedlings, the reads primarily represent 24mers from both strands at similar frequencies. These 24mers are absent in *Pl-Rh* cobs and seedlings indicating the change in sRNA abundance between allele states largely represents changes in 24mer production. Interestingly, in cob profiles, we noted an antisense-specific bias for 23mers that may represent a strand bias for RNAP IV transcription [76]. The *Pl-Rh / Pl´* heterozygotes also produced USR 24mers at levels similar to those of *Pl´* individuals indicating that the formerly *Pl-Rh* allele may already be a source of 24nt RNA biogenesis in day-8 seedlings. In contrast, 4cm cob sRNA libraries from the B73 inbred had few reads representing the corresponding repeat region in the B73 genome (see S7 Fig).

We next previewed the entire penta-repeat unit sRNA landscapes by including both unique and multi-mapping reads from *Pl-Rh*, *Pl´* and *Pl-Rh / Pl´* in cob and seedling (Methods). In all datasets, profiles were dominated by a similar multimapping read cluster precisely over the repetitive DNA-7_ZM TE, dwarfing the adjacent USR read peak (see S8 Fig). While the genomic source of these TE sRNAs is unknown, they could contribute to recruiting RNAP IV or V in the *Pl´* state.

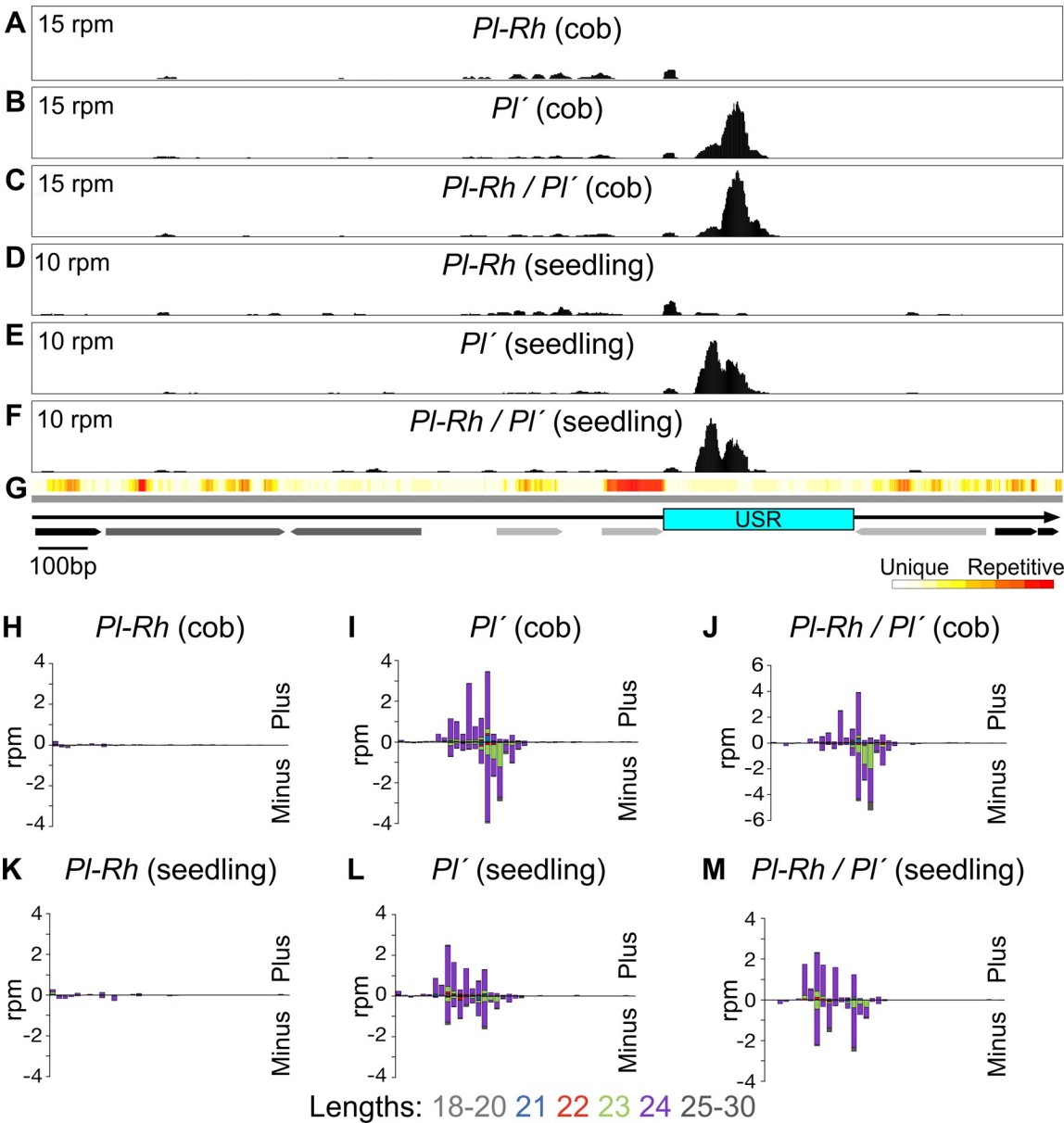

**Fig 6. Penta-repeat sRNA profiles.** Alignments of uniquely-mapping 18-30nt reads from libraries representing single *Pl-Rh* (**A**), *Pl´* (**B**), and *Pl-Rh / Pl´* (**C**) immature cob and seedling (**D**-**F**) across a single repeat unit (**G**) in reads per million (rpm). Arrows represent DNA transposons (light gray), *Helitrons* (black), and LTR retrotransposons (dark gray). Mean rpm over the USR in 10nt bins is shown in stacked bar graphs for cob *Pl-Rh* (**H**), *Pl´* (**I**), and *Pl-Rh / Pl´* (**J**), and seedling *Pl-Rh* (**K**), *Pl´* (**L**), and *Pl-Rh / Pl´* (**M**) genotypes. Reads colored by length mapping to the plus or minus stands are shown above and below the line respectively.

We expected the USR 24mers should be sourced by RNAP IV because RPD1 is responsible for nearly all maize 24nt RNA production [34]. To test this, we compared sRNA reads from ~4cm cob libraries representing two combined *Pl´; Rpd1 / rpd1-1*, and one *rpd1-1 / rpd1-1* sibling sample (Fig 7A and 7B). Because 24nt RNAs are largely depleted in *rpd1-1* mutants, other size classes are relatively more abundant in sRNA datasets; to compensate, we normalized the fraction of each size class in both mutant and non-mutants over the relative fraction of 21nt RNAs which are unaffected by loss of RNAP IV [34]. Less than 10% of genome-wide 24mers

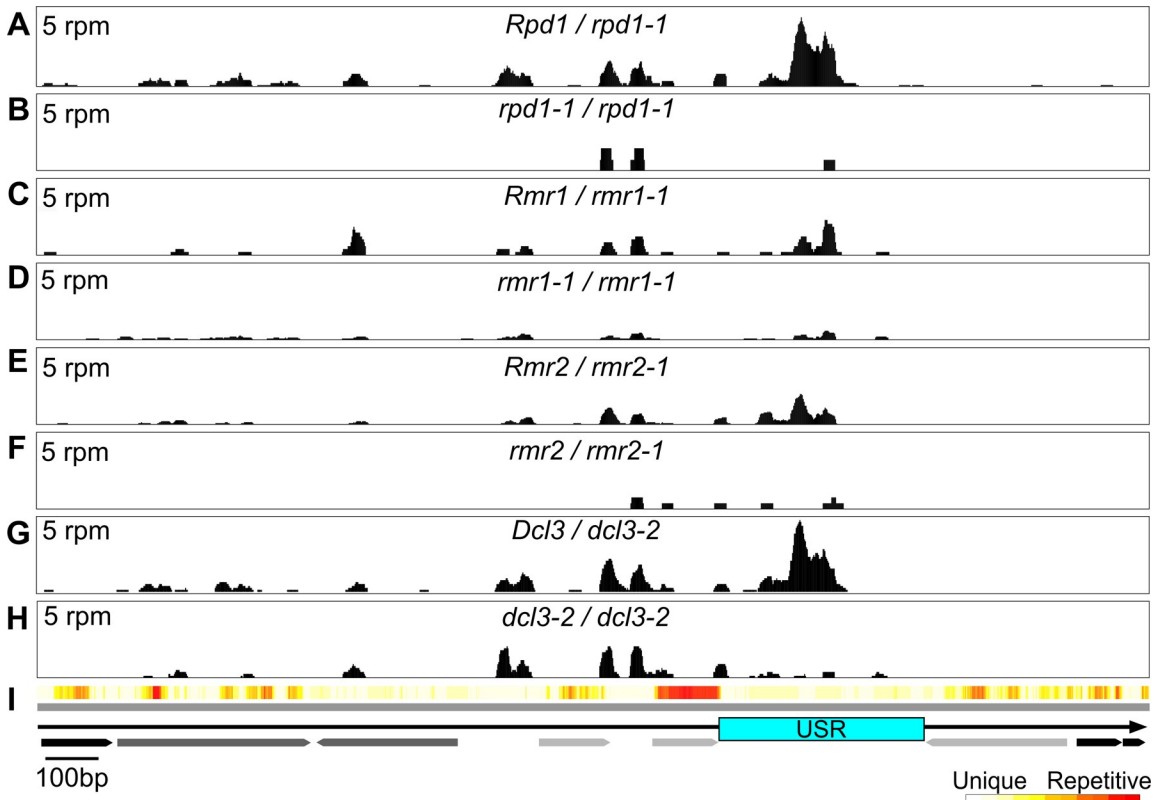

**Fig 7. Unique penta-repeat sRNA profiles dependent on RMR functions.** Alignments of uniquely-mapping 18-30nt reads from libraries representing *Rpd1 / rpd1-1* (two combined) (**A**), *rpd1-1 / rpd1-1* (**B**), *Rmr1 / rmr1-1* (**C**), *rmr1-1 / rmr1-1* (**D**), *Rmr2 / rmr2-1* (**E**), *rmr2-1 / rmr2-1* (**F**), *Dcl3 / dcl3-2* (**G**), and *dcl3-2 / dcl3-2* (**H**) immature cob across a single repeat unit (**I**) in reads per million (rpm). Arrows represent DNA transposons (light gray), *Helitrons* (black), and LTR retrotransposons (dark gray).

remained in the *rpd1-1* mutant library (see S7 Table) and few USR sRNA reads of any size were represented (Fig 7B). We therefore conclude that the majority of USR sRNAs depend on RNAP IV, though the specific RNAP IV subtype(s) acting there remains unknown.

Peptides of RMR1 and its paralog CHR167 co-purify with epitope-tagged RPD1 [37], therefore it is possible that these two putative helicases define additional functional RNAP IV subtypes to those distinguished by alternate second largest subunits. Because ethidium-bromide staining of gel-fractionated sRNAs indicated that RMR1 is required for ~68% of all immature cob 24nt RNAs [32], we compared ~4cm cob sRNA profiles from single *Pl´*; *Rmr1 / rmr1-1* and *rmr1-1 / rmr1-1* siblings. In contrast to the *rpd1* mutant profiles, in the *rmr1-1* mutant, genome-wide normalized 24mers (see S7 Table; see Methods) were ~48% of those of their heterozygous sib, and these reductions were almost exclusive to repetitively-mapping 24mers. Nearly all USR sRNA reads were depleted in the *rmr1-1* mutant library (Fig 7C and 7D), indicating that one or more penta-repeat USRs are targeted by an RMR1-specific RNAP IV subtype in this reproductive tissue.

We also reanalyzed existing 4cm cob sRNA datasets from *rmr2-1* and *dcl3-3* mutants and heterozygous non-mutant siblings that indicated RMR2 and DCL3 account for ~65% and ~95% of all 24mers, respectively [39,41]. Abundant USR-derived sRNA reads were found in both non-mutants but were depleted in the corresponding *rmr2-1* and *dcl3-3* mutant libraries (Fig 7E–7H). Thus, in all the *rmr* mutants we profiled, there is a loss of USR-derived 24mers, even when abundant genome-wide 24mers remained in *rmr1-1* (48%) and *rmr2-1* (35%)

mutants. These findings highlight a subset of RNAP IV-dependent 24mers whose USR-specific biogenesis is associated with paramutagenic *Pl´* states.

## USR cytosine methylation patterns distinguish *Pl-Rh* and *Pl´* states

Since *Pl-Rh* and *Pl´* are distinguished by differential USR sRNA production, we examined whether these also correspond with differing 5mC patterns. Seedling-derived genomic DNA samples isolated from single isogenic *Pl-Rh*, *Pl´*, and *Pl-Rh / Pl´* individuals were treated with APOBEC (NEB) to deaminate unmethylated cytosines and then the USR segment where differences in sRNA production and MNase-sensitivity were observed was PCR amplified. Resulting amplicons were then cloned and Sanger-sequenced. We observed that this USR segment in *Pl-Rh* has almost no 5mC in any context while ~88% of cytosines in CG and 55–70% in CHG–yet few (6–7%) in CHH–contexts were methylated in both *Pl´* and *Pl-Rh / Pl´* genotypes (Fig 8). These data show that reduced expression and paramutagenic action of the *Pl´* state correlate with elevated 5mC levels at most, if not all, of the five USR copies.

## Discussion

Our findings are consistent with the *Pl1-Rhoades* penta-repeat acting as both a distal transcriptional enhancer and as a feature required for paramutagenic function. The only abundance difference of uniquely mapping sRNAs distinguishing paramutable *Pl-Rh* and paramutagenic *Pl´* across 192kb occurs at one or more USR copies within the penta-repeat. When *Pl1-Rhoades* is in the highly expressed *Pl-Rh* state, portions of the USR display differential nuclease sensitivity, increased polyadenylated RNA levels, few 5mCs, and physical association with sequences immediately upstream of the *Pl1-Rhoades* TSS. From these findings, we conclude that the USR has regulatory significance for the *Pl1-Rhoades* haplotype. Furthermore, these data indicate that the USR is RNAP II-transcribed in the *Pl-Rh* state but RNAP IV-transcribed in the *Pl´* state. Thus, this small non-coding region within the penta-repeat context appears to serve as a switchable distal element specifying meiotically heritable regulatory states.

Although challenges remain in defining features of active plant enhancers, chromatin accessibility, histone acetylation, absence of 5mC, and promoter proximity appear diagnostic [77]. Both the *b1* hepta-repeat and proximal *P1-rr* repeated sequences can promote transcriptional activation in transgenesis-based reporter assays [58,60]. Compared to the repressed *B´* state, the *b1* hepta-repeat sequences in the active *B-I* state 1) are more DNase I sensitive, 2) are more represented in a FAIRE-based protein-depleted DNA fraction, 3) are associated with elevated H3 acetylation [51,78], 4) have reduced 5mC levels [51], and 5) physically associate with the *b1* CDS at higher frequencies [79]. The hepta-repeat junction sequences are also, like metazoan enhancers (reviewed in [80–82]), bi-directionally RNAP II-transcribed, but this occurs at similar levels in both the *B-I* and *B´* states [47]. Likewise, sRNA levels so far appear similar across the hepta-repeat [38,47] whereas the abundance of both penta-repeat polyadenylated transcripts and RNAP IV-dependent sRNAs distinguish *Pl-Rh* from *Pl´*. The *P1-rr* enhancer is also transcribed at similar rates between allele states, but like *Pl1-Rhoades*, sRNA levels appear higher in the paramutant state [50]. These distinct characteristics point to variations in the mechanism driving paramutation at different loci; this intriguing finding demands further investigation.

The polyadenylated USR-containing RNAs associated with the active *Pl-Rh* state indicate this region is transcribed, but the origin(s), direction(s), and length(s) of these RNAs remains unknown. Because neither Arabidopsis RNAP IV nor RNAP V produce polyadenylated RNAs [83–87], we infer that these USR-containing transcripts are RNAP II-derived. Whether this transcription reflects a more open chromatin nucleosome-free configuration region, however,

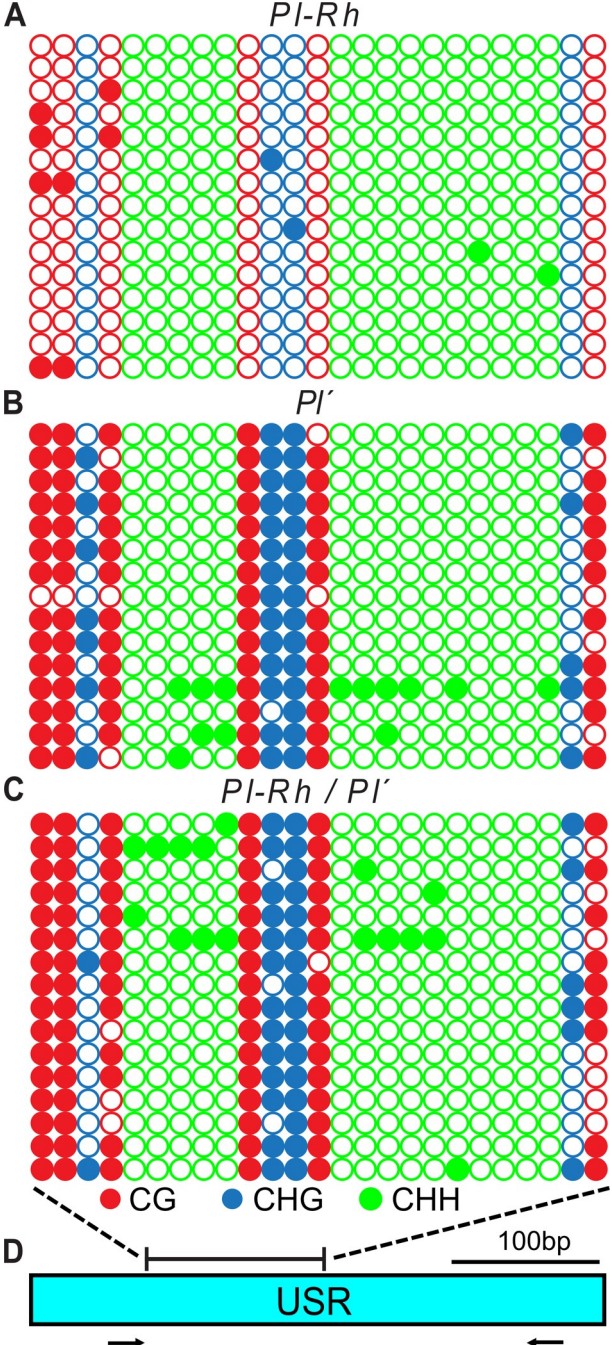

**Fig 8. USR 5mC profiles.** Cytosine methylation profiles of individual PCR amplicons of genomic DNA from *Pl-Rh* (**A**), *Pl´* (**B**) and *Pl-Rh / Pl´* (**C**) day-8 seedlings representing the bracketed region of the USR shown in (**D**). Circles (open: unmethylated, closed: methylated) represent cytosines in CG (red), CHG (blue), and CHH (green) contexts. Black arrows in (**D**) represent PCR primers used to generate the sequenced amplicons.

remains equivocal. Because the B73 AGPv4 annotated PAS of the adjacent gene immediately downstream of the penta-repeat is less than 100bp from the end of the last repeat, one idea is that pre-termination transcription continues into the repeat array and a downstream PAS is used. Alternatively, the penta-repeat RNAs could initiate within USR sequences or from its

resident TE fragments. In either case, a nascent RNA containing USR sequences and/or the adjacent highly repetitive DNA-7_ZM TE could serve as an essential scaffold transcript for the initial recruitment of RNAP V via sRNA-directed argonaute complexes [88]. Hence, a paramutable *Pl-Rh* state may require such polyadenylated penta-repeat RNAs for paramutation initiation while a paramutagenic *Pl´* state likely uses RNAP V-sourced transcripts to maintain sRNA biogenesis.

The relationship of sRNAs with endogenous alleles undergoing paramutation has to this point remained unclear. Here we found that the appearance of USR-specific 24nt RNAs in both early seedlings and immature cobs precisely correlates with the reduced plant pigment phenotypes and paramutagenic action diagnostic of *Pl´* states [6]. The similar USR sRNA profiles observed between *Pl´* and *Pl-Rh* / *Pl´* seedlings indicate that RNAP IV is likely already recruited to the naive *Pl-Rh* allele state early in development. Our mutant sRNA analyses show that loss of sRNA biogenesis from a discrete USR segment is coincident with increased *Pl1-Rhoades* transcription (*rmr2-1*, *rpd1-1*, and *dcl3-3*), increased *Pl1-Rhoades* mRNA-stability (*rmr1-1*) and, meiotically-heritable reversion of *Pl´* to *Pl-Rh* states (*rmr1-1*, *rmr2-1*, *rpd1-1* and *dcl3-3*) [31,32,39,41,74].

Because 24nt USR RNA production depends on RNAP IV and other RMR proteins in an RdDM-like mechanism, it is reasonable to posit that RNAP IV-dependent chromatin changes at the USR impede *Pl1-Rhoades* transcription. Given that *rmr1-1* mutant cobs lose most USR-specific 24mers, this indicates that the RNAP IV subtype occupying the USR in immature cobs is associated with RMR1 and not its CHR167 paralog [37]. The redistribution of sRNAs seen in *chd3a* mutants argues that CHD3a specifies where the seedling-based RNAP IV complexes function. Since CHD3 proteins are associated with nucleosome remodeling [89], and the Arabidopsis CHD3 PICKLE can move nucleosomes *in vitro* [90], it is likely that the *chd3a-3* mutant penta-repeat sRNA profile reflects changes in nucleosome positioning that impact RNAP IV localization. The appearance of 5mC, coincident with USR sRNA biogenesis, agrees with a model in which CHD3a-positioned nucleosomes direct RNAP IV-dependent 24nt RNA production that, analogous to Arabidopsis RdDM, specifies *de novo* 5mC patterns.

We found a correlation between 5mC appearance at the USR sequences and the *Pl´* state. 5mC presence appears generally associated with alleles displaying paramutagenic behaviors, including 1) at the 5´ ends of divergently oriented *r1* genes in *R-r:standard* repressed states [48], 2) within repeats flanking the *p1* CDS in paramutagenic *p1* haplotypes [5,44,49,58], 3) at the *b1* hepta-repeat in the *B´* state–with the exception of specific cytosines being more methylated in *B-I* [51,59,60,78], 4) the repressed phenotype conferred by a paramutagenic *lpa1-241* allele is partially compensated by 5-azacytidine treatment [7] implicating 5mC in repression, and 5) in Arabidopsis autotetraploids, a spontaneously silenced epiallele of a partially-repeated hygromycin resistance transgene displays paramutagenic properties coincident with increased 5mC and 24mers primarily across the repeat region [14,91]. Moreover, 6) in tomato, the paramutagenic *sulfurea* allele also has increased 5mC in the promoter region and part of the 5´ CDS [92]. Changes in 5mC and sRNA production are also common when divergent genomes are combined [9,93,94], but whether these are causative of, or responsive to, changes in gene regulation remains unknown. Although 5mC presence is a unifying hallmark of paramutagenic allele states, 5mC alone is insufficient to predict paramutagenicity.

Arabidopsis RdDM facilitates *de novo* cytosine methylation in CG, CHG, and CHH contexts (reviewed in [33,95]), and once established, these modifications can be maintained in the absence of RdDM by a maintenance methyltransferase (MET1; CG) and a pair of chromomethyltransferases (CMT3; CHG, and CMT2; CHH) [96] that are recruited to methylated H3K9 [97]. Because maize lacks a CMT2 ortholog [96], mCHH presence is assumed to reflect ongoing RdDM [98]. Interestingly, mCHH is specifically found in an enhancer region of the

paramutagenic *P1-rr´* allele, but not in the stably silenced, non-paramutagenic, *P1-pr^{TP}* allele [44]. Indeed, maize has little genome-wide mCHH (~5%); its presence is more correlated with sRNA production than with mCG or mCHG [94], and loss of MOP1/ZmRDR2, RP(D/E)2a, or RPD1 results in genome-wide reductions [98]. Although most maize mCHH depends on active RdDM, the CMT3 orthologs encoded by *zmet2* and *zmet5* are required to maintain some mCHH [98]. Since most of our sequenced USR amplicons from *Pl´* and *Pl-Rh / Pl´* seedlings are devoid of mCHH, we conclude that active RdDM is not occurring in most of the nuclei we sampled. Therefore, the high mCG (88%) and mCHG (55–70%) levels we observed mostly reflect maintenance methylation. While initial recruitment and *trans*-homolog communication of paramutation may depend on an RdDM-like mechanism primarily occurring in cells not sampled by our analysis, 5mC maintenance pathways are likely required for somatic and intergenerational stability of repressed allele states.

While 5mC could be a meiotically heritable feature specifying paramutagenicity, it remains curious that, so far, no downstream (effector) proteins in an RdDM-like pathway have been identified in the *mop-* and *rmr*-based genetic screens. In Arabidopsis, sRNAs may direct the action of the SUVH2 and SUVH9 lysine methyltransferases to specify H3K9me2 [99], which can be maintained in a feedforward loop with mCHG through the actions of CMT3 and SUVH4/5/6 [100]. Arabidopsis SAWADEE HOMEODOMAIN HOMOLOGUE1 (SHH1) recruits RNAP IV to H3K9me2 [101]. A related maize ortholog, ZmSHH2a, also binds methylated H3K9 with the highest affinity for H3K9me1 [102]. ZmSHH2a is complexed with RP(D/E)2a and MOP1/ZmRDR2 [37], therefore Wang et al. [102] propose that ZmSHH2a forms a connection between RNAP IV recruitment and H3K9me1. Thus, similar to eukaryotes such as fission yeast that lack any cytosine methylation machinery, H3K9me could represent a heritable mark as well [103]. Indeed, loss of either tomato CMT3 or SUVH4 orthologs lead to reactivation of a paramutagenic *sulfurea* allele [104]. Additionally, because CHD3 proteins affect the regulation of loci marked by H3K27me3 [105–107], and CHD3a influences penta-repeat 24nt RNA production, it is possible that H3K27 modifications are also involved in maintaining a meiotically heritable epigenetic mark distinguishing repressed *Pl1-Rhoades* states. Loss of MOP1/ZmRDR2 or RPD1 results in a reduction of both H3K9 and H3K27 dimethylation at the *b1* hepta-repeat, suggesting a connection between these marks and *b1* paramutation [78]. Indeed, recent work in *Drosophila* implicates H3K27me3 and inter-chromosomal contacts in facilitating paramutation at the Fab2L transgene [19].

Gametic sRNAs may also contribute to the inheritance of repressed *Pl´*. In other metazoan examples of paramutation, it is the cytoplasmic transmission of Argonaute-bound sRNAs that confers heritable paramutagenic action (reviewed in [25]). Indeed, Argonaute-bound sRNAs are sufficient to initiate RdDM in Arabidopsis [88] and shRNA constructs can induce the change of *B-I* to *B´* in transgenic maize [47]. In maize, however, paramutagenicity exclusively co-segregates with specific alleles [4,6,58,108]. Additionally, when aneuploid sperm cells are generated from haploid *Pl´* gametophytes–as can be accomplished using B-A chromosomes [109]–only sperm cells contributing *Pl´* chromosomes transmit paramutagenic function [110]. This result indicates that any sperm-transmitted sRNAs are by themselves insufficient to induce paramutation.

We found that, similar to specific *r1* and *b1* haplotypes, repeat copy numbers correlate with paramutagenic activities [57,59,61]. Importantly, the presence of one or two USRs in non-paramutable *pl1* haplotypes indicates that more repeats are required to obtain paramutagenic properties. In *r1* paramutation, repeated *r1* coding regions and flanking sequences need not be adjacent, or in *cis*, to promote paramutagenic function [61]. Because the NC350 and CML52 haplotypes share identical 3´ repeats, we expect that either the solo repeat found upstream of *Pl1a-CML52* and/or the additional *Pl1b-CML52* gene confers its paramutagenic properties. In

addition, the *pl1-bol3* haplotype containing three *pl1* gene copies–none having the *doppia* fragment [111]–is reportedly paramutagenic [112]. The *pl1-bol3* downstream region remains uncharacterized, however, so whether any paramutagenic activity is dependent on *pl1* gene copy number or other *cis*-regulatory features is unclear. Our findings are consistent with the emerging consensus that repeated sequences are essential for paramutagenic function.

The distinct sequences and structural arrangements of the *B1-I* and *Pl1-Rh* distal repeats may be related to the observed differences in allele stabilities and locus-specific RMR requirements. Neither RMR1 nor RMR12 repress *B'* [32,40]. Each individual unit of the *b1* heptarepeat (853bp) consists entirely of unique sequence [59], while the 2092bp penta-repeat unit is mainly consisting of TE fragments with an embedded 390bp unique sequence. Unlike *B'*, where heritable reversions to a fully expressed *B-I* state have not been observed [52,113,114], the *Pl'* state is less stable and can revert to *Pl-Rh* when either hemizygous [115], heterozygous with certain recessive alleles [66,115] or in the persistent absence of RMR1 [31,41], RPD1 [74], CHD3a [40], or DCL3 [39]. Whether the size, copy number, location, and/or composition of these *b1* and *pl1* repeats contribute to the observed differences in allele behaviors and mechanistic requirements remains to be explored.

Depending on the phenotypic trait affected, metastable alleles having paramutation-like properties may be desirable or detrimental to breeding efforts. With long-read-based whole genome assemblies now resolving repeated sequences, it should be possible to predict which alleles of genes having agronomic importance might show similar dynamic behaviors. Such efforts will require ATAC-seq profiles and Hi-C interaction maps to identify repeat-associated enhancers, and sRNA distributions to highlight regions of ongoing RdDM. Additionally, RNA-seq profiles of RNAP IV-deficient plants would help validate the regulatory significance of such repeats. Crop improvement programs can now identify and manage these examples of heritable epigenetic variation.

## Methods

### Genetic materials

Inbred lines A619, A632, and B73 were sourced from the North Central Regional Plant Introduction Station (USDA-ARS, Ames, IA). The *Pl1-Rhoades* haplotype introgressed into A619 ([74]; herein designated A619 *Pl1-Rhoades*; 99.25% A619 for *Pl-Rh* and 98.5% A619 for *Pl'*) came from a previously described color-converted W23 stock (herein designated W23 *Pl1-Rhoades*) [6] developed by Ed Coe, Jr. (USDA-ARS, University of Missouri, Columbia, MO) and maintained in several lines obtained from the Maize Genetics Cooperation Stock Center (MGCSC)(USDA-ARS, University of Illinois, Urbana, IL) and Vicki Chandler (University of Oregon, Eugene, OR). A *T6-9 (043–1)* [116] interchange chromosome (*T Pl1-Rhoades*) having linkage between *Pl1-Rhoades* (*6L*) and a recessive *9S waxy1* allele (*wx1*) was introgressed into B73 (herein designated B73 *T Pl1-Rhoades*; 98.44%) as previously described [66,74] and into A619 (herein designated A619 *T Pl1-Rhoades*; ~97% A619) and A632 (herein designated A632 *T Pl1-Rhoades*; 99.25% A632). The BAC library and 4C analysis were derived from *Pl-Rh* type A619 *Pl1-Rhoades* plants. qRT-PCR was performed on W23 *Pl1-Rhoades*, A619 *Pl1-Rhoades*, and B73 *T Pl1-Rhoades* individuals. MNase assays were done with A619 *Pl1-Rhoades* seedlings. 5mC analyses and sRNA libraries were made from isogenic homozygous *Pl-Rh*, *Pl'* and heterozygous (*Pl-Rh / Pl'*) cobs and seedlings of B73 *T Pl1-Rhoades* parentage (progenies 161453, 161483, 161482, 170671, 170673, 170674). Additional sRNA libraries were derived from immature cobs of *Rpd1/rpd1-1* and *rpd1-1/rpd1-1* siblings from progeny 80031 (~25% W23, 25% A632) and *Rmr1/rmr1-1* and *rmr1-1/rmr1-1* siblings from progeny 80886 in A632 (BC$_4$F$_3$). For mutant analyses, all non-mutant siblings

were homozygous for *Pl1-Rhoades* in a *Pl´* state while mutants had darky colored anthers consistent with derepression of *Pl1-Rhoades*.

## BAC identification and sequencing

Snap-frozen developing cobs (progenies 91353 and 91363) were submitted to Amplicon Express (Pullman, WA) to construct a ~3.7X coverage pCC1 BAC-based library and representative Hybond-N+ (Amersham Biosciences) filter arrays. 73,728 clones were screened by formamide-based hybridization to a random-primed 32P radiolabeled 0.9kb *Hin*dIII fragment from the pJH1 plasmid [54]. BAC DNA from ten clones was isolated using a Qiagen miniprep kit and evaluated for *pl1* sequences using *pl1*-specific PCR amplification (see S8 Table for primers). Six PCR-positive clones (60_P21, 25_M18, 51_M22, 53_O24, 161_K19, 192_C12) were cultured with Epicentre Biotechnologies (Madison, WI) CopyControl BAC Induction Solution and BAC DNA was isolated with modification to an established protocol [117] including 1μl of 100mg/ml RNaseA in Solution I and adding a final phenol:chloroform extraction followed by chloroform extraction and isopropanol precipitation. BAC ends were Sanger sequenced by the Vincent J. Coates Genome Sequencing Laboratory (UC Berkeley) as per Poulsen and Johnson [118] (see S8 Table for primers) using 2μg of BAC template and 3 pmol of primer, in 20μl Sanger sequencing reactions with the following thermocycle steps: 3 min at 96˚C followed by 99 cycles of 10 sec at 96˚C, 10 sec at 50˚C, 4 min at 60˚C. BLASTn aligned the respective end sequences of all six BACs to *6L* positions flanking the *pl1* gene model. All six BAC clones were later prepared and sequenced on the Pacific Biosciences (PacBio) platform at The Genome Analysis Centre, TGAC, Norwich, UK (Now Earlham Institute).

## BAC assembly and annotation

Raw PacBio reads (see S1 Table for statistics) were filtered and assembled into partial or complete linear contigs at The Genomic Analysis Centre (see S3 File). BAC backbone sequences were identified by BLASTn and removed. The resulting genomic sequences were aligned to each other and the existing *Pl1-Rhoades* reference using the *Multiple Align* feature of *Geneious* (v6.1.8; [119]). Together, these BAC sequences extended the *Pl1-Rhoades* reference sequence to 192kb, with six independent BACs supporting the sequence over the *pl1* coding region and four supporting the sequence over the penta-repeat. One of the BAC sequences (60_P21) was unable to be assembled into a single contig, so only those fragments which overlap the consensus sequence of the other BAC sequences were included. A fragment that only overlapped the consensus by 1kb consisting entirely of *Gypsy* transposon sequence was also excluded. Gene models were predicted by BLASTing the assembled BAC sequence to cDNA molecules from the B73 AGPv4 genome [70]. Transposons and TE fragments were predicted using CENSOR on April 26, 2022 ([71]; see S4 File).

## Haplotype comparisons

~70kb extracted from the *Pl1-Rhoades* BAC contig containing the *Pl1-Rhoades* allele and surrounding region was aligned to the corresponding region in the B73 AGPv4 genome [70] and the draft CML52 and NC350 genomes [73] with *GEvo* [120] using the default settings with a minimum high-scoring segment pair size of 400nt. The resulting alignments were manually redrawn using Adobe Illustrator at scale.

## 4C analysis

Tissue was collected from biological triplicate inner husk leaves at silking and from inner stem leaves from A619 *Pl1-Rhoades* (*Pl-Rh*) V5 seedlings. Interacting chromatin was isolated from these tissues as described for 3C experiments [79,121,122]. In short, 2-3g tissue was crosslinked in 1X PBS with 2% formaldehyde for 1 hour. Nuclei were resuspended in 1.2X Buffer B (Roche) and treated with 0.2% (w/v) SDS followed by 2% (w/v) Triton X-100 and digested with 400 U *Dpn*II (NEB, 50U/μl) overnight. *Dpn*II was heat-inactivated, and for intra-molecular ligation of crosslinked digested DNA, the sample was incubated in a volume of 7ml with 100U T4 ligase (Promega, 20U/μl) at 16˚C for 5 hours, followed by 45 min at RT. Chromatin was de-crosslinked and ligated DNA (3C sample) was recovered by precipitation and the pellet dissolved in 150μl of 10mM Tris-HCl pH 7.5. To generate 4C samples, the 3C samples were digested and ligated essentially as described [123]. The DNA was digested with 50U *Nla*III (NEB, 5U/μl), *Nla*III was heat-inactivated and the digested DNA was re-ligated overnight in a total volume of 14ml as described above. The ligated DNA was precipitated, dissolved and column purified (QIAquick PCR purification kit- Qiagen). To produce 4C libraries, for each sample, 8 PCR reactions were performed. 100ng of 4C DNA served as template in each inverse PCR reaction. PCR was conducted with the Expand Long Template PCR System (Roche, #11681834001) using primers fused to P5 (forward) and P7 (reverse) illumina sequencing adapters; a forward primer immediately upstream of the *Pl1-Rhoades* CDS, and a reverse primer at the 5´ end of the *doppia* transposon fragment upstream of *Pl1-Rhoades* facing away from the gene (see S8 Table). The program used was 2min 94˚C; [30sec 94˚C, 30sec 52˚C, 2min 72˚C] 35x; 10min 72˚C, 12˚C ∞. Proper 4C amplicons contained: P5 - forward primer (bait sequence) - *Dpn*II - interacting sequence - *Nla*III - reverse primer - P7. The 8 PCR reactions per sample were pooled and purified with the Qiaquick PCR purification kit (Qiagen). Library concentrations were measured using the Nanodrop spectrophotometer, mixed in equal concentrations and sequenced on an Illumina 2000 platform by BGI Tech Solutions (Hongkong) Co., producing single-end 49 bp reads. Replicates were processed in different lanes.

Reads were first filtered to those containing the *Dpn*II cutsite and the 3 preceding nt of the forward primer to ensure the *Dpn*II site was in the correct context using the *BBTools* (https://jgi.doe.gov/dataand-tools/bbtools/) function *bbduk* (options: k = 7 mink = 7 hdist = 1 adapter: CTCGATC). The resulting reads were then trimmed up to the *Dpn*II cutsite using *bbduk* (options: ktrim = l k = 16 mink = 12 hdist = 2 maxlen = 28, adapter: TACGCCGGCG-GAGCTC). Since each read is 49bp, setting the maximum length to 28 ensured that any read where the forward primer could not be recognized due to sequencing errors would not be retained. The *Nla*III site and reverse primer, if present, were then trimmed using *bbduk* (options: ktrim = r k = 12 mink = 7 hdist = 1 minlen = 8, adapter: CATGAGCTATGA). Setting the minimum length to 8 ensured the reads contained an interacting sequence, and eliminated potential primer dimers; reads with longer interacting sequences were not expected to contain the *Nla*III site, so presence of this site was not required. Lastly the reads were quality filtered and trimmed using *bbduk* (options: qtrim = r maq = 20 minlen = 8). After these filtering steps, the third replicate from each tissue had significantly fewer reads than the other two libraries (see S3 Table) and these were excluded from further analyses. The remaining four libraries were aligned to the B73 AGPv4 genome [70] using *Bowtie* (v0.12.8; options: -v 0 --best -m 1 -S) [124]. Those reads which either mapped uniquely or did not map were aligned to a collapsed version of the *Pl1-Rhoades* BAC-based sequence containing only one copy of the penta-repeat, allowing no mismatches or multiply -mapping reads, using *Bowtie* (v0.12.8; options: -v 0 --best -m 1 -S) [124]. All clean reads (not filtered to the genome) were also aligned to the

assembled BAC sequences as above. All alignments were collapsed to their 5´ most coordinate and converted to BED format using *awk*. The abundance of tags at each position of the BAC sequence was determined using *BEDTools* genomecov [125]. Only positions with counts were retained, and 2kb on either side of the bait region were excluded to avoid artifacts. The resulting counts were normalized to rpm and ranked by total abundance (see S4 and S5 Tables).

## Chromatin profiling

Mono and dinucleosome-bound DNA was obtained by isolating nuclei from six pooled day-8 A619 *Pl1-Rhoades* (*Pl-Rh*) and (*Pl´*) [74] seedlings in triplicate and fractionating MNase-treated chromatin as described [126]. A range of MNase concentrations was tested to identify a mild treatment which produced a full ladder representing different numbers of nucleosomes bound to DNA following gel electrophoresis. An amount of nuclei yielding ~20μg DNA were incubated for 4 minutes at 37°C with 20U MNase (Thermo Scientific EN0181) in MNase reaction buffer (50mM Tris pH 7.5, 320mM sucrose, 4mM $MgCl_2$, 1mM $CaCl_2$) before proteinase K treatment and DNA extraction as described [126]. The bands corresponding to mono and dinucleosomes were extracted from the gel, pooled, and three technical replicate qPCR reactions were performed on three biological replicates. Abundance of each USR amplicon was normalized to abundance of *alanine aminotransferase* (*alt4*; Zm00001d014258) (see S8 Table for primers). DNA was amplified with SensiMix SYBR No ROX (Bioline), and data was collected using an Eppendorf Mastercycler EP Gradient S thermocycler. Cycle threshold (Ct) values were calculated using the *noiseband* option in Eppendorf Mastercycler EP Realplex V2.2 software.

## qRT-PCR

To quantify *pl1* mRNA, total RNA was isolated from biological duplicates of *Pl-Rh* and *Pl´* inner seedling and husk leaves from both A619 *Pl1-Rhoades* and B73 *T Pl1-Rhoades* lines. cDNA was generated, and qRT-PCR was performed using an Applied Biosystems 7500 Real-Time PCR System on technical duplicates. *pl1* mRNAs were normalized to those from *actin1* (Zm00001d010159) (see S8 Table). To quantify USR transcripts, total RNA was isolated from male flowers (florets) of triplicate *Pl-Rh* and *Pl´* type W23 *Pl1-Rhoades* individuals [6] just prior to anthesis. Three florets from a single individual were pooled per sample. RNA was isolated, cDNA was generated, qRT-PCR was performed, and fold expression was calculated as described [40] except the cycle threshold (Ct) values were established using the *CalQplex* option in Eppendorf Mastercycler EP Realplex V2.2 software. Candidate enhancer transcripts were amplified with P3_F and P3_R primers (see S8 Table) and normalized to *gapdh* which was amplified using previously published primers ([127]; see S8 Table). Raw Ct values are provided in S5 File.

## sRNA sequencing

Biological triplicate day-8 seedling and 4cm immature cob B73 *T Pl1-Rhoades* (*Pl-Rh*), (*Pl´*) and (*Pl-Rh / Pl´*) sRNA libraries (SRA accessions SRR15400896-SRR15400912) were made from TRizol-extracted total RNA using the Bioo Scientific (Austin, TX) Nextflex Small RNA-Seq Kit v3 as described [40], and 150bp paired-end sequencing was performed by Novogene Co. Ltd. on an Illumina HiSeq4000 platform. Only two seedling *Pl-Rh / Pl´* libraries were successfully sequenced. Between 37M and 198M reads were produced per library (see S6 Table for library statistics). Two B73, two *Rpd1 / rpd1-1*, one *rpd1-1 / rpd1-1*, one *Rmr1 / rmr1-1*, and one *rmr1-1 / rmr1-1* 4cm immature cob libraries (GEO GSE52103 and SRA SRR15410418-SRR15410422) were prepared as described [41]. Briefly, TRizol-extracted total

RNA were separated by PAGE, sRNAs were gel-extracted, and libraries were prepared according to the Illumina sRNA Sequencing Guide. Adapters were trimmed from *rmr* mutant and heterozygous cob libraries, deduplicated, and quantified using a *Perl* script. B73 cob libraries were filtered and trimmed using the *BBTools* (https://jgi.doe.gov/dataand-tools/bbtools/) function *bbduk* (options: ktrim = r k = 18 mink = 11 hdist = 1 maq = 10 minlen = 18 maxlength = 30); B73 immature cob libraries were not deduplicated. *Pl-Rh*, *Pl´*, *Pl-Rh / Pl´* and previously published *Chd3a / Chd3a* and *chd3a-3 / chd3a-3* ([40]; GEO GSE158990) raw reads were filtered using *bbduk* with options (ktrim = r k = 18 mink = 11 hdist = 1 minlen = 26 maxlength = 38). Only mate 1 was used from paired-end libraries. The resulting trimmed reads were converted to .fasta format using *seqtk* (https://github.com/lh3/seqtk) and deduplicated using the *FASTX-Toolkit* command *fastx_collapser* (http://hannonlab.cshl.edu/fastx_toolkit/index.html). The Nextflex Small RNA-Seq Kit v3 adapters have 4N tags flanking the insert. These tags were removed using *seqtk* (options: trimfq -b 4 -e 4). *bbduk* was used to filter *rpd1*, *rmr1*, and *rmr2* immature cob datasets by length (options: minlen = 18 maxlen = 30). *rpd1* and *rmr1* whole genome mapping statistics (see S7 Table) were obtained as previously described for *rmr2* and *dcl3* [39,41]. Percent sRNAs of each size class were then normalized by the ratio of the percent 21mers in the mutants to the percent 21mers the corresponding heterozygotes since 21mer levels appear unchanged in these mutants [34,69].

18-30nt reads from all datasets were aligned to known maize rRNA and tRNA sequences (provided by Dr. Blake Meyers, Donald Danforth Plant Sciences Center) using *Bowtie* (v0.12.8; options: -v 0 --best -m 1 -S) [124]. Those sequences not mapping to tRNA/rRNA were considered "clean" and aligned to the B73 AGPv4 genome [70] using *Bowtie* (v0.12.8; options: -v 0 --best -m 1 -S) [124]. Those reads (excluding the B73 libraries) that either mapped uniquely or did not map were aligned using *ShortStack* [75] to a modified version of the *Pl1-Rhoades* haplotype sequence with the five repeats collapsed to a single copy to allow reads which could map to any repeat to map uniquely. *ShortStack* clusters were called using the six libraries from *Pl-Rh* and *Pl´* cobs since these were the deepest datasets (*ShortStack* options: -- mismatches 0 --mmap n --nohp --mincov 152 --pad 50), and sRNAs from all other libraries were quantified at these defined loci using the *ShortStack* option --locifile. Minimum coverage of 152 represents 0.5 rpm for those datasets, and the non-default pad of 50 (minimum distance between independent clusters) was chosen to prevent *ShortStack* from artificially collapsing independent clusters into a single locus because of the depth of these libraries. Reads mapping uniquely to each nucleotide of the composite unique subregion were collapsed to their 5´ most nucleotide and separated by strand and length using *awk*, quantified using the *samtools* function *samtools-depth* [128], and normalized to rpm. Normalized counts are displayed in 10nt bins (Fig 6H–6M). In parallel, all clean reads from *Pl-Rh*, *Pl´*, and *Pl-Rh / Pl´* were mapped to the collapsed BAC sequence using *Bowtie* (v1.3.1; options: -v 0 --best). Representative alignments are shown for each tissue and genotype in S8 Fig.

## 5mC profiling

Genomic DNA was isolated from B73 *T Pl1-Rhoades* (*Pl-Rh*), (*Pl´*) and (*Pl-Rh / Pl´*) aerial day-8 seedling tissue. ~160ng DNA was mixed with 30ng unmethylated lambda DNA and sheared using a Branson sonicator (two 15 sec pulses at 7watts with 45 sec rest), and unmethylated cytosines were converted to uracils using the NEBNext Enzymatic Methyl-seq Conversion Module (NEB #E7125) followed by PCR using standard *Taq* DNA polymerase (NEB)(see S8 Table) to amplify fragments within the USR and lambda control. Lambda_F and R were used on the *Pl-Rh* and *Pl´* samples while Lambda_F2 and R2 were used on *Pl-Rh / Pl´* sample. The PCR products were then purified using AxyPrep Mag PCR Clean-Up Kit (Axygen

MAG-PCR-CL-5). All PCR amplicons - with exception of the lambda amplicon from the *Pl-Rh / Pl′* sample - were cloned into pGEM-T easy vectors (Promega A1360) and transformed into competent NEB5-alpha *E. coli* (NEB #C2987H). Colonies were selected by blue white screening and purified plasmid DNA was Sanger sequenced at The Genomics Shared Resource in the OSU Comprehensive Cancer Center. A 123bp region with full coverage high quality sequence in all amplicons was analyzed by Kismeth [129] for cytosine methylation. For the *Pl-Rh / Pl′* sample, Sanger sequencing was performed on the lambda amplicons directly. Complete cytosine conversions were seen for all lambda controls (see S9 Table).

## Supporting information

**S1 Fig. Structures of *pl1* haplotypes.** (**A**) Structure of a single copy of the *Pl1-Rhoades* penta-repeat. Cyan box: unique subregion (USR) used as a hybridization probe. Arrows represent DNA transposons (light gray), *Helitrons* (black), and LTR retrotransposons (dark gray). *Bgl*II (B) and *Nsi*I (N) restriction sites producing hybridizing fragments in the *Pl1-Rhoades* (**B**) and *pl1-B73* haplotypes (**C**) are shown. Black boxes represent gene models, large arrows represent the penta-repeat sequences, cyan boxes represent the USR, and green shading represents the region of recombination for *pl1-R30*. (**D**) Southern blot of *Pl1-Rhoades* (*Pl1-Rh*), *pl1-B73*, and *pl1-R30* genomic DNA digested with *Bgl*II and *Nsi*I and probed with a radiolabeled USR fragment.
(PDF)

**S2 Fig. *Pl1-Rhoades* mRNA expression profiles.** Mean fold *pl1* mRNA ($2^{-\Delta\Delta Ct}$) in *Pl-Rh* and *Pl′* inner seedling and husk leaves relative to inner husk leaves from *Pl′* individuals measured by qRT-PCR normalized to *actin1* levels in biological duplicate plants from A619 and B73 lines.
(PDF)

**S3 Fig. sRNA clusters across the *Pl1-Rhoades* haplotype.** (**A**) Locations of 30 clusters called by ShortStack across the *Pl1-Rhoades* haplotype (blue bars) with clusters 24 and 25 which overlap the penta-repeat (**B**) highlighted. Arrows represent DNA transposons (light gray), *Helitrons* (black), and LTR retrotransposons (dark gray).
(PDF)

**S4 Fig. Immature cob *doppia* sRNA profiles.** Alignments of uniquely-mapping 18-30nt reads from libraries representing single *Pl-Rh* (**A-C**), and *Pl′* (**D-F**) immature cobs across the *doppia* fragment upstream of the *Pl1-Rhoades* coding sequence and the 5' region of *Pl1-Rhoades* exon 1 (**G**) in reads per million (rpm). (**H**) Clusters called by ShortStack.
(PDF)

**S5 Fig. Immature cob penta-repeat sRNA profiles.** Alignments of uniquely-mapping 18-30nt reads from libraries representing single *Pl-Rh* (**A-C**), *Pl′* (**D-F**), and *Pl-Rh / Pl′* (**G-I**) immature cobs across a single repeat unit (**J**) in reads per million (rpm). Arrows represent DNA transposons (light gray), *Helitrons* (black), and LTR retrotransposons (dark gray).
(PDF)

**S6 Fig. Seedling penta-repeat sRNA profiles.** Alignments of uniquely-mapping 18-30nt reads from libraries representing single *Pl-Rh* (**A-C**), *Pl′* (**D-F**), and *Pl-Rh / Pl′* (**G-H**) seedlings across a single repeat unit (**I**) in reads per million (rpm). Arrows represent DNA transposons (light gray), *Helitrons* (black), and LTR retrotransposons (dark gray).
(PDF)

**S7 Fig. B73 sRNA profiles across penta-repeat region.** Alignments of uniquely-mapping 18-30nt reads from libraries representing single B73 inbred (**A-B**) immature cobs across B73 sequence (**C**) equivalent to the *Pl1-Rhoades* penta-repeat repeat unit (**D**) in reads per million (rpm) with structural differences highlighted (dotted lines). Arrows represent DNA transposons (light gray), *Helitrons* (black), and LTR retrotransposons (dark gray). Only transposons unique to B73 are labeled in (**C**).
(PDF)

**S8 Fig. Multimapping penta-repeat sRNA profiles.** Alignments of all unique and multimapping 18-30nt reads from libraries representing single *Pl-Rh*, *Pl′*, and *Pl-Rh* / *Pl′* cob (**A-C**) and seedlings (**D-F**) across a single repeat unit (**G**) in reads per million (rpm). Arrows represent DNA transposons (light gray), *Helitrons* (black), and LTR retrotransposons (dark gray).
(PDF)

**S1 Table. BAC sequencing statistics.**
(DOCX)

**S2 Table. Paramutagenic properties of *pl1* haplotypes from NAM founder lines.**
(DOCX)

**S3 Table. Library statistics for 4C datasets.**
(XLSX)

**S4 Table. 4C tag abundances (all).**
(XLSX)

**S5 Table. 4C tag abundances (unique).**
(XLSX)

**S6 Table. Library statistics for sRNA datasets.**
(DOCX)

**S7 Table. *rmr* mutant sRNA genome mapping statistics.**
(XLSX)

**S8 Table. Primer sequences.**
(DOCX)

**S9 Table. Enzymatic cytosine conversion efficiency of unmethylated lambda DNA.**
(DOCX)

**S1 Methods. Paramutagenicity tests.**
(PDF)

**S1 File. Fasta-formatted *Pl1-Rhoades* haplotype sequence.**
(TXT)

**S2 File. sRNA library statistics.**
(XLSX)

**S3 File. Fasta-formatted BAC sequences containing *Pl1-Rhoades*.**
(TXT)

**S4 File. CENSOR-based, gff3-formated, transposon annotations.**
(TXT)

**S5 File. Raw Ct values for qPCR experiments.**
(XLSX)

## Acknowledgments

We thank the Maize Genetics Cooperation Stock Center (MGCSC)(USDA-ARS, University of Illinois, Urbana, IL) and North Central Regional Plant Introduction Station (USDA-ARS, Ames, IA) for contributing valuable genetic materials. We are grateful to Drs. Blake Meyers (Donald Danforth Plant Sciences Center) and Stacey Simon (University of Delaware) for constructing and sequencing the immature cob sRNA libraries from the various *rmr* mutants and non-mutant siblings. We thank Dr Mario Caccamo (NAIB East Malling Research) for his assistance in sequencing the *Pl1-Rhoades* BACs (sequencing was delivered via the BBSRC National Capability in Genomics (BB/J010375/1) at the Earlham Institute by members of the Genomics Pipelines Group), Drs. Pawel Krajewski and Dimitrios Zisis (Institute of Plant Genetics, Polish Academy of Sciences) for advice with the 4C data analysis, and Ludek Tikovsky and Harold Lemereis for taking care of the maize plants (Swammerdam Institute for Life sciences, University of Amsterdam). NCBI, Gramene, and MaizeGDB provided critical sequence and data resources. Maize nurseries were supported in part by the University of California College of Natural Resources Oxford Facilities Unit, the Ohio Agricultural Research and Development Centers Waterman Agricultural and Natural Resources Laboratory, and The Ohio State University College of Arts & Sciences Biological Sciences Greenhouse.

## Author Contributions

**Conceptualization:** Natalie C. Deans, Joy-El R. B. Talbot, Iris Hövel, Darren Heavens, Maike Stam, Jay B. Hollick.

**Data curation:** Natalie C. Deans, Joy-El R. B. Talbot, Iris Hövel, Darren Heavens, Maike Stam.

**Formal analysis:** Natalie C. Deans, Joy-El R. B. Talbot, Mowei Li, Cristian Sáez-González, Iris Hövel, Darren Heavens, Jay B. Hollick.

**Funding acquisition:** Maike Stam, Jay B. Hollick.

**Investigation:** Natalie C. Deans, Joy-El R. B. Talbot, Mowei Li, Cristian Sáez-González, Iris Hövel, Darren Heavens, Jay B. Hollick.

**Methodology:** Natalie C. Deans, Joy-El R. B. Talbot, Mowei Li.

**Project administration:** Maike Stam, Jay B. Hollick.

**Resources:** Darren Heavens, Maike Stam, Jay B. Hollick.

**Software:** Natalie C. Deans, Joy-El R. B. Talbot, Darren Heavens.

**Supervision:** Maike Stam, Jay B. Hollick.

**Validation:** Natalie C. Deans, Joy-El R. B. Talbot, Jay B. Hollick.

**Visualization:** Natalie C. Deans, Joy-El R. B. Talbot, Mowei Li, Jay B. Hollick.

**Writing – original draft:** Natalie C. Deans, Jay B. Hollick.

**Writing – review & editing:** Natalie C. Deans, Joy-El R. B. Talbot, Mowei Li, Cristian Sáez-González, Iris Hövel, Darren Heavens, Maike Stam, Jay B. Hollick.

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
