## [Decision Letter · Decision Letter 0]

8 Feb 2024

Dear Dr Hollick, dear Jay,

Thank you very much for submitting your Research Article entitled 'Paramutation at the maize *pl1* locus is associated with RdDM activity at distal tandem repeats' to PLOS Genetics. Please let me apologize for the delay of the response, and thanks for the patience.

The manuscript was now fully evaluated at the editorial level and by three independent peer reviewers. As you will see from the detailed comments, they all value the careful and meaningful work by which you document important aspects of the paramutation at the pl1 locus, provide a lot of data on the components involved and compare it with similar cases in plants and animals. I agree with the reviewers that, though some aspects resemble the situation at other paramutation loci in maize, the differences are interesting and deserve to be pointed out. In addition to some minor suggestions for improved figures or legends, there are some points that need a bit more attention. I am aware that some of the major suggestions, though reasonable, would require substantial experimental extension, and they might not be essential for the publication. Other main points are easier to address, by extending the data analysis, minor additional work, or good discussions. I therefore suggest that you go through the comments and provide the requested information, extend the Discussion, or address the reviewers’ comments otherwise. We will require a detailed list of your responses to the review comments and a description of the changes you have made in the manuscript.

If you decide to revise the manuscript for further consideration at PLOS Genetics, please aim to resubmit within the next 60 days, unless it will take extra time to address the concerns of the reviewers, in which case we would appreciate an expected resubmission date by email to plosgenetics@plos.org.

We are sorry that we cannot be more positive about your manuscript at this stage. Please do not hesitate to contact us if you have any concerns or questions.

Best regards, Ortrun

Ortrun Mittelsten Scheid

Academic Editor

PLOS Genetics

John Greally

Section Editor

PLOS Genetics

As you will see from the detailed comments of the three reviewers, they all value the careful and meaningful work by which you document important aspects of the paramutation at the pl1 locus, provide a lot of data on the components involved and compare it with similar cases in plants and animals. I agree with the reviewers that, though some aspects resemble the situation at other paramutation loci in maize, the differences are interesting and deserve to be pointed out. In addition to some minor suggestions for improved figures or legends, there are some points that need a bit more attention. I am aware that some of the suggestions, though reasonable, would require substantial experimental extensions, and I do not see them as essential for the publication. Other main points are easier to address, by extending the data analysis, minor additional work, or good discussions of the points. I therefore suggest that you go through the comments and provide the requested information, extend the Discussion, or answer directly to the reviewers’ comments.

Reviewer's Responses to Questions

**Comments to the Authors:**

Reviewer #1: This manuscript focuses on identification and characterization of sequences required for pl1 paramutation. This is a large manuscript, with multiple sets of genetic and molecular data, that summarizes efforts of many researchers.

Briefly, authors identified a sequence required for pl1 paramutation. This sequence is located downstream of the pl1 coding sequence and exists as an array of three to five repeats in alleles that participate in paramutation. Alleles that do not participate in paramutation have one or two repeats. Each repeat is ~2 kb in length and there is a smaller (389 bp) unique sequence (USR) embedded in each ~2kb repeat. USR exhibits following molecular characteristics; it produces polyA transcripts, accumulates low levels of 24 nt sRNAs, and exhibits differential DNA methylation. Based on 4C data, which shows an association of the penta-repeat array with the 5’ of the pl1 gene, authors propose that penta-repeats function as a pl1 enhancer. Manuscript also details analyses of sRNA accumulation in multiple mutant backgrounds and authors extensively discuss the role of RdDM in pl1 paramutation. While there are some pl1 paramutation-specific molecular characteristics, majority of findings are consistent with those previously reported for paramutation at r1, b1 and p1 loci.

The similarities to other cases of paramutation in maize in no way negate the value of this manuscript. Uncovering and documenting similar trends between different cases of paramutation is essential for forming a comprehensive understanding of how paramutation is regulated. Collectively, this and prior studies demonstrated that repeated enhancer arrays carrying unique locus-specific DNA are the key feature of the phenomenon of paramutation. Based on this, I think that the data presented in this paper are of interest to a broad circle of researchers working in the area of epigenetics. I recommend this manuscript for publication, after minor issues detailed below are addressed.

Minor comments:

Page 24, line 428, says “paramutagenic pl1 haplotypes…”. Should it be “paramutagenic p1 haplotypes”? The references that follow this phrase are for p1, not pl1 locus.

Page 27, Lines 487-497. It would help if it was made clearer how this paragraph relates to the experimental data in the manuscript. It does not seem to add much to the discussion.

Figure 2. Two suggestions here:1) What “Total” in panel A is showing? Because of the Y axis scale, it does not look as it shows the sum of panels B and C. Is it the average of panels B and C? Please, clarify in the figure legend. 2) I suggest adding tracer lines to connect the expanded diagram in panel E to the corresponding section of DNA in panel D. This would help with faster comprehension of the figure.

Figure 8. I propose adding tracer lines to connect diagram of methylation sites in panels A-C to the location of the sequence assayed in panel D (bracketed black line below the USR box).

S. Fig 3. I propose to add tracer lines that would indicate which region of sequence in panel A is expanded in panel B. This modification would make it easier to understand the figure.

Reviewer #2: Deans et al., report on the characterization of a distal cis-regulatory element involved in paramutation at the pl1 locus that is located approximately 14 kb downstream of the pl1 polyadenilation site and is formed by five copies of a repetitive fragment organized as a tandem repeat. The authors present a thorough and comprehensive review of the current knowledge on paramutation in plants that is mechanistically contrasted in the discussion with information coming from animal systems (this was a pleasant addition). There is a lot of work included in this manuscript. The authors performed chromatin conformation analysis indicating that the penta repeat comes in close proximity to the promoter of pl1. Analysis of sRNA-seq data in Pl-Rh an Pl’ show differential accumulation of sRNAs mapping to the penta repeat in two different tissues (cob and seedlings). Analysis of the expression pattern of sRNAs mapping to the penta repeat in mutant backgrounds of the RNA-directed DNA methylation pathway (RdDM) genes indicate a role for classical RdDM components including small RNAs (sRNAs), NRPD1 (POL4), DICER-LIKE 3 (DCL3 ) and other non-classical such as CHROMODOMAIN HELICASE DNA-BINDING 3 (CHD3). Taken together, the authors present a model (based on the model proposed for paramutation at the b1 locus).

The data is interesting since the genetic requirements for paramutation at pl1 are different relative to paramutation at the b1 locus, including the molecular nature of the tandem repeats and the function of genes that are unique to pl1 for example CHD3 is not required for paramutation at the b1 locus. The authors propose that paramutation at the pl1 locus involves a mechanism based on RdDM or an RdDM-like mechanism that invokes transcription by RNA POL II in Pl-Rh and transcription by RNA POL IV in Pl’. Does the observation that in Arabidopsis POLIV-dependent transcripts lack a 3’-poly(A) tail has been shown to be the same in maize?. In Arabidopsis POLIV-dependent transcripts are produced from both strands. Did you test this at the USR in Pl-Rh and Pl’?

Given there is no experimental evidence supporting a role for CHD3 in positioning RNA POLIV, What prevents POL4 from transcribing in Pl-Rh?. CHD3 is not required for paramutation at the b1 locus so what is special about the penta repeat in pl1 that requires non-general components such as CHD3 that might be interacting with RNA POLIV? Does the POLIV isoform that is proposed to interact with CHD3 also includes MOP2/RMR7 or a different second largest subunit?

The accumulation pattern of the sRNAs mapping to the USR in the same tissue (immatue cobs) across the different mutant backgrounds makes it difficult to understand the actual role of these sRNAs. Paramutation is not affected in heterozygous Rmr1 / rmr1-1 and Rmr2 / rmr2-1 individuals. However there is a clear reduction (around 50% of maybe more) in the levels of sRNAs mapping to the USR in Rmr1 / rmr1-1 and Rmr2 / rmr2-1. How does the model account for this? Do the differences in 5mC observed in Pl-Rh and Pl’ seedlings are the same in wild-type cobs and cobs from Rmr1 / rmr1-1 and Rmr2 / rmr2-1 heterozygous individuals? What is the accumulation pattern of sRNAs mapping to the USR in cobs from Chd3a / chd3a-3 individuals?

Since loss-of-function of ZmDCL3 results in a modest effect on global 5mC levels and sRNAs mapping to the USR are dramatically depleted in dcl3-2 / dcl3-2. What are the 5mC levels at the USR in the dcl3-2 / dcl3-2 mutant background? What is the accumulation pattern of sRNAs mapping to the USR and the 5mC levels at the USR in a MOP2/RMR7 mutant background?

Reviewer #3: In this manuscript Deans et al. convincingly identify the regulatory region for maize pl1 paramutation: distal tandem repeats likely acting as enhancers. The reported similarities and differences with other loci susceptible to paramutation in maize and other species are an important step to better understand the mechanism of this epigenetic phenomenon. The manuscript is very clear and the data well presented. I have found no major issues with the current manuscript and therefore only provide suggestions for improvements

suggestions:

chromatin conformation: Could you perform 4C for the Pl’ allele? It would be interesting to see whether the silenced form of the tandem repeats still interacts with the pl1 promoter. A reciprocal 4C from the viewpoint of the USR would also inform on the potential for the tandem repeat to affect other genes in the region, and check those genes’ expression for potential paramutation.

sRNA profiles: it would be helpful to also present the alignment results with the multi-mapping reads, to better understand what the full population of sRNAs at this locus looks like. Possibly on a Pl1-Rh haplotype-substituted B73reference. Can you comment on the observed 24 nt RNA levels compared to other clusters genome-wide?It doesn’t look like they are super abundant, which is compatible with the mostly symmetric mC and the fact that paramutation loci aren’t predicted by massive amounts of 24nt sRNAs (at least in the tissues studied)

I am intrigued by the USR Pol II-dependent RNA: is it some kind of long non-coding RNA?Can public RNA-seq data be mined to infer its boundaries?Can you comment on its absolute expression level? For instance if it’s basically 0 in Pl’, is a 15-fold increase in Pl-Rh still practically 0? Full-length cDNA + capture + long-read sequencing could also help resolve what’s being transcribed there.

DNA methylation: a higher-throughput method would be great. If there are 5 repeats, the 10 Sanger reads could all be from the same nucleus in theory! At a minimum, you could directly sequence the PCR product. If DNA input is not limiting, I’d recommend Cas9-targeted nanopore sequencing, you could get 100X depth, encompass the whole tandem array and find interesting patterns from repeat to repeat, and you can add guide RNAs for other interesting regions like the pl1 promoter in the same experiment. The public long-read nanopore datasets might also enable DNA methylation analysis for this region in B73 at least.

l. 168 add “in convergent orientation” to make it clear the tandem repeat is between the two genes.

**Have all data underlying the figures and results presented in the manuscript been provided?**

Reviewer #1: Yes

Reviewer #2:&n

---

## [Decision Letter · Decision Letter 1]

8 May 2024

Dear Dr Hollick, Dear Jay,

We are pleased to inform you that your manuscript entitled "Paramutation at the maize *pl1* locus is associated with RdDM activity at distal tandem repeats" has been editorially accepted for publication in PLOS Genetics. All reviewers appreciated the consideration of their comments and the revised version. Congratulations!

Best regards, Ortrun

Ortrun Mittelsten Scheid

Academic Editor

PLOS Genetics

John Greally

Section Editor

PLOS Genetics

Comments from the reviewers (if applicable):

Reviewer's Responses to Questions

**Comments to the Authors:**

Reviewer #1: Comments and suggestions that I made in my review have been addressed in the revised manuscript. I have no further comments or suggestions. I recommend this manuscript for publication.

Reviewer #2: Congratulations to the authors for addressing all the questions and putting together a very nice and exciting story.

Reviewer #3: Thank you for taking my comments into account. I fully support publication.

You’re right that Nanopore on non-CG methylation wasn’t very good (DeepSignal-plant was ok, though not friendly to run) but the current all context model (R10.4, v3.0, 5mC_5hmC v1) seems better.

I look forward to reading the follow-up studies.

**Have all data underlying the figures and results presented in the manuscript been provided?**

Reviewer #1: None

Reviewer #2: Yes

Reviewer #3: Yes

PLOS authors have the option to publish the peer review history of their article (what does this mean?). If published, this will include your full peer review and any attached files.

Reviewer #1: No

Reviewer #2: No

Reviewer #3: **Yes: **Quentin Gouil

**Data Deposition**

http://datadryad.org/submit?journalID=pgenetics&manu=PGENETICS-D-24-00023R1

**Press Queries**

---

## [Editor Report · Acceptance letter]

24 May 2024

PGENETICS-D-24-00023R1 

Paramutation at the maize *pl1* locus is associated with RdDM activity at distal tandem repeats 

Dear Dr Hollick, 

We are pleased to inform you that your manuscript entitled "Paramutation at the maize *pl1* locus is associated with RdDM activity at distal tandem repeats" has been formally accepted for publication in PLOS Genetics! Your manuscript is now with our production department and you will be notified of the publication date in due course.

With kind regards,

Anita Estes

PLOS Genetics

On behalf of:
